# Inhibiting ACK1-mediated phosphorylation of C-terminal Src kinase counteracts prostate cancer immune checkpoint blockade resistance

Dhivya Sridaran[1,2,13], Surbhi Chouhan[1,2], Kiran Mahajan[1,2,3,13], Arun Renganathan [1,2], Cody Weimholt[1,4,5], Shambhavi Bhagwat[1,2], Melissa Reimers[3,4], Eric H. Kim[1,2], Manish K. Thakur[6], Muhammad A. Saeed [3], Russell K. Pachynski[3,4], Markus A. Seeliger [6,7], W. Todd Miller[7,8], Felix Y. Feng [9,10,11,12] & Nupam P. Mahajan [1,2,3] ✉

Solid tumours are highly refractory to immune checkpoint blockade (ICB) therapies due to the functional impairment of effector T cells and their inefficient trafficking to tumours. T-cell activation is negatively regulated by C-terminal Src kinase (CSK); however, the exact mechanism remains unknown. Here we show that the conserved oncogenic tyrosine kinase Activated CDC42 kinase 1 (ACK1) is able to phosphorylate CSK at Tyrosine 18 (pY18), which enhances CSK function, constraining T-cell activation. Mice deficient in the *Tnk2* gene encoding Ack1, are characterized by diminished CSK Y18-phosphorylation and spontaneous activation of CD8[+] and CD4[+] T cells, resulting in inhibited growth of transplanted ICB-resistant tumours. Furthermore, ICB treatment of castration-resistant prostate cancer (CRPC) patients results in re-activation of ACK1/pY18-CSK signalling, confirming the involvement of this pathway in ICB insensitivity. An ACK1 small-molecule inhibitor, (*R*)-**9b**, recapitulates inhibition of ICB-resistant tumours, which provides evidence for ACK1 enzymatic activity playing a pivotal role in generating ICB resistance. Overall, our study identifies an important mechanism of ICB resistance and holds potential for expanding the scope of ICB therapy to tumours that are currently unresponsive.

The initiation of T-cell antigen receptor (TCR) signalling is dependent on the coordinated activation of a series of SRC-family tyrosine kinases, including CSK[1–3]. The target of CSK is LCK, a key kinase; LCK activity is precisely regulated by two distinct tyrosine phosphorylation events with opposite consequences. LCK Y394 phosphorylation in the activation loop is needed to acquire full kinase activity; in contrast, Y505 phosphorylation in the C-terminal tail, which interacts with its own SH2 domain and promotes a closed autoinhibition conformation, causes loss of kinase activity. The crucial role played by CSK was determined using transgenic mice expressing a mutant form of CSK (CskAS), which showed reduced CSK activation and led to enhanced immune responses to very weak antigens, including antigens that might not have activated T cells under normal circumstances[4]. These data suggest that targeting CSK activation may be a promising strategy for reactivating the immune response. Despite the importance of CSK activation, our

---

understanding of the regulation of CSK activity during TCR signal initiation remains inadequate.

Metastatic castration-resistant prostate cancer (CRPC) is a malignancy that poses a major challenge due to its immunosuppressive tumour microenvironment. As the second leading cause of cancer deaths in American men[5], CRPCs have attracted considerable attention by researchers seeking to address resistance mechanisms. T-cell exhaustion due to chronic tumour antigen stimulation can be successfully overcome via immune checkpoint blockade (ICB) targeting cytotoxic T-lymphocyte antigen-4 (CTLA-4), programmed cell death-1 (PD-1), or programmed cell death ligand-1 (PD-L1)[6,7]. Although ICBs have emerged as major successes against certain cancers[7], prostate cancer has been found to be highly refractory to ICBs[8], exhibiting marginal efficacy in clinical trials when administered either as a single agent or in combination with other agents[9]. Metastatic CRPC patients with a high intratumoral CD8+ T-cell density and an IFN-γ response gene signature exhibited favourable responses to CTLA-4-targeting antibodies[10], suggesting that therapeutic strategies that increase CD8+ T-cell activation may show clinical benefits in CRPCs. However, targeted activation of CD8+ T cells in advanced prostate tumours using a therapeutically amenable small-molecule compound has not yet been achieved.

We previously revealed that a nonreceptor tyrosine kinase, ACK1 (also known TNK2)[11,12], acts as an epigenetic kinase and deposits novel pY88-H4 epigenetic marks on the androgen receptor (AR) enhancer[13]. ACK1 is a multidomain protein (Supplementary Fig. 1a) that is highly expressed in the spleen and thymus[14,15]. Multiple malignancies favour ACK1 overexpression; ACK1 gene amplification, mutations, and aberrant kinase activation have been reported in various malignancies, including lung, breast, ovarian and prostate cancer[12,13,16–24]. ACK1 levels are tightly regulated in cells; seven in absentia homologue (SIAH) ubiquitin ligases bind the degron motif located in the C-terminus of ACK1, driving its proteasomal turnover[25].

The rationale of this study was to examine the role of ACK1 in immune regulation. We observe ACK1 activation in two antagonizing cell types, immune cells and cancer cells, suggesting a dual mode of ACK1 action. T-cell priming and T-cell trafficking to the tumour microenvironment are dampened by activated ACK1. Consistently, genetic ablation or pharmacological inhibition of ACK1 impairs tumour growth by activating T cells and promoting persistent immune surveillance. The implication of ACK1 inhibition is evident in reactivating an immune response in ICB-resistant tumours, opening a new therapeutic modality for 'cold' tumors.

## Results

### Genetic ablation of *Tnk2* lowers the TCR response threshold and amplifies T-cell responsiveness

To explore the role played by ACK1 kinase, we generated a viable *Tnk2* conditional knockout (Ack1 KO here onwards) mouse model (Supplementary Fig. 1b; see Methods section). *Ack1*flx/wt mice were bred with EIIa-Cre mice to determine whether loss of *Ack1* leads to embryonic lethality. The adenovirus EIIa promoter directed the expression of Cre recombinase in preimplanted mouse embryos, and thus, loss of Ack1 expression was expected in nearly all tissues. *Ack1* heterozygous mice were generated, and they were subsequently interbred to obtain homozygous KO mice (Supplementary Fig. 1c, d). Whole-body knockdown of *Ack1* was confirmed by immunoblotting and qRT–PCR (Fig. 1a–c). No embryonic lethality was observed in the *Ack1*-KO mice. The *Ack1*-KO mice were similar to wild-type (WT) mice in terms of general physical appearance and did not show any physical disability (Supplementary Fig. 1d). Since ACK1 is primarily expressed in the spleen and thymus, we reasoned that the loss of ACK1 may influence the immune system, and therefore, we performed an extensive analysis of the splenocytes by staining them with antibodies against activation markers. The percentages of CD44hiCD62Llow memory CD8+ T cells and CD137+ effector CD8+ T cells were increased in the spleen of

*Ack1*-KO mice, albeit no differences were observed in the ratios of Foxp3−CD4+ helper T, Foxp3+CD4+ regulatory T cells (Tregs), CD8+ CTLs, CD19+ B, NK1.1+ NK, and CD11b Gr-1+ myeloid cells (Fig. 1d, e and Supplementary Figs. 2 and 3).

T-cell activation upon anti-CD3 antibody treatment resulted in significant reduction in ACK1 Y284 phosphorylation and LCK Y505 phosphorylation; in contrast, a significant increase in LCK Y394 phosphorylation was observed, suggesting that loss of ACK1 activation coincided with T-cell activation (Supplementary Fig. 4a). Furthermore, anti-CD3 antibody treatment caused an increase in the CSK levels (Supplementary Fig. 4b). To examine the consequences of the loss of Ack1 on the ability of T cells to induce calcium mobilization, we examined calcium flux in splenocytes obtained from WT and KO mice. Calcium influx is one of the crucial events during T-cell receptor (TCR) engagement and subsequent signalling that induces T-cell activation[15,26–28]. Splenocytes obtained from KO mice exhibited an increased calcium response compared to those obtained from WT mice (Fig. 1f). Furthermore, splenocytes from WT and KO cells exhibited a significant increase in calcium flux upon anti-CD3 antibody treatment (Fig. 1g). Moreover, increases in blood and splenocyte IFN-γ levels (Fig. 1h and Supplementary Fig. 4c) and IL-2 levels (Fig. 1i and Supplementary Fig. 4d) were observed. Thus, the genetic ablation of *Ack1* endowed immune cells with faster and larger calcium responses and may play a role in T-cell responsiveness.

The increase in the basal T-cell response shows the potential to cause tissue damage and may contribute to the development of autoimmune diseases. We assessed whether there were any traces of autoimmunity in organs such as the liver, kidney, small intestine, and lungs using histology and picrosirius red staining. We also assessed liver function by performing an alanine transaminase (ALT) assay, and we measured anti-nuclear antibodies (ANA) in the blood serum of WT and KO mice. ALT assays of the WT and KO mice samples revealed no difference and indicated an absence of tissue damage (Supplementary Fig. 4e). Similarly, there was no change in the levels of ANA between the WT and KO mice (Supplementary Fig. 4f). Moreover, histological analysis of the organs stained with H&E and picrosirius red from older WT and KO mice showed minimal signs of inflammation or fibrosis (Supplementary Fig. 4g, h). Taken together, these data indicate that, although it activated the T-cell response, ACK1 kinase loss does not lead to a hyperreactive response or generate autoimmune disease. However, the possibility that *Ack1*-KO mouse-derived T cells may be prone to mount an immune response against autoantigens after viral infection or other immune stresses cannot be ruled out.

### ACK1 phosphorylates CSK at Y18, restraining LCK activation

To decipher the mechanistic details of ACK1 kinase signalling, total Tyr-phosphorylated proteins were immunopurified from ACK1-expressing cells and subjected to mass spectrometry-based identification of posttranslational modifications. The phosphorylation of many known ACK1 substrates, such as histone H4 (pY88)[13], Wiskott–Aldrich syndrome protein (pY256)[29], and NCK (pY112 and 251)[30], and multiple autophosphorylation sites in ACK1 (pY284, 992, 193, 40, 859, 860, 232, 242, and 423) were identified. In addition, we observed previously unknown phosphorylation of Tyr at the 18th position in the SH3 domain of CSK (Supplementary Fig. 5a). This global phosphoproteomic screen also revealed three distinct phosphorylation sites, pY227, pY317, and pY417, in the CSK-binding protein (Cbp), also known as PAG (Supplementary Fig. 5b–d). The Y18 site has been evolutionarily conserved in all Src-family tyrosine kinases (SFKs) except LCK (Supplementary Fig. 5e), suggesting that ACK1 may target CSK but not LCK, forming a linear signalling nexus, ACK1-CSK-LCK.

In previous work, we have shown that ACK1 interacts with the SH3 domains in SFKs, such as SRC and HCK, via the ACK1 C-terminal proline-rich domain[31]. We reasoned that the C-terminal proline-rich domain of ACK1 may also be involved in CSK interaction. To examine

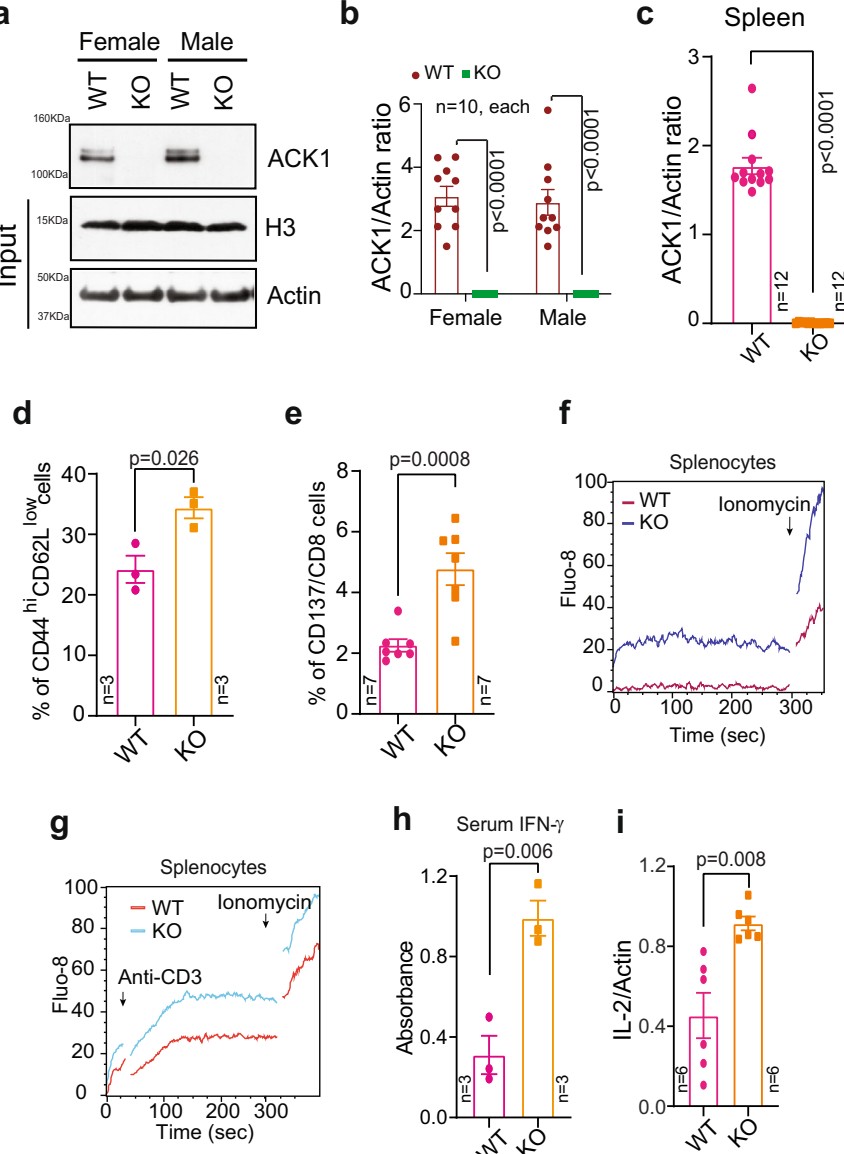

**Fig. 1 | Genetic ablation of *Ack1* endows mouse primary T cells with enhanced T cell responsiveness. a** Brain lysates from WT and *Ack1* KO mice were immuno-blotted with ACK1 antibody. H3 and Actin were used as input controls. **b** RNA was isolated from the brain of male and female WT and *Ack1* KO mice, followed by qRT-PCR with *Ack1* and actin primers ($n = 10$ mice in each group, 3 replicates). **c** RNA was isolated from the spleen of WT and *Ack1* KO mice, followed by qRT-PCR with ACK1 and actin primers ($n = 12$ mice in each group, 3 replicates). **d** Flow cytometric analysis of splenocytes isolated from WT and *Ack1* KO mice to assess CD44$^{hi}$CD62L$^{low}$ expression on CD8 gated population ($n = 3$ mice in each group). **e** T cells were purified from splenocytes of WT and *Ack1* KO mice and flow cyto-metry was performed to assess the expression of CD137 on CD8 gated population ($n = 7$ mice in each group). **f** A representative calcium flux of splenocytes from WT

and *Ack1* KO mice. The cells were loaded with the calcium indicator dye Fluo-8, and Ionomycin (1 µg/ml) was added at the 300th sec. Data are representative of three independent mice experiments. **g** A representative calcium flux of splenocytes from WT and *Ack1* KO mice. The cells were loaded with the calcium indicator dye Fluo-8, and anti-CD3 antibody (0.5 µg/ml) was added at 30th sec. Ionomycin was added at the 300th sec. Data is representative of three independent mice experiments. **h** Blood serum levels of IFN-γ in WT and *Ack1* KO mice determined by ELISA. ($n = 3$ mice in each group, 3 replicates). **i** RNA was prepared from spleen of WT and *Ack1* KO mice, followed by real time RT-PCR of IL-2. ($n = 6$ biologically independent samples, 3 replicates). For **b**–**e**, **h** and **i**, the data are represented as mean ± SEM from three independent experiments and *p* values were determined by unpaired two-tailed Student's *t*-test. Source data are provided as a Source Data file.

this possibility, full-length ACK1 and cACK1 deletion constructs carrying an intact and active kinase domain but not a proline-rich domain (Supplementary Fig. 1a) were coexpressed with CSK. Coimmunoprecipitation studies revealed that ACK1 bound and phosphorylated CSK; in contrast, cACK1 not only failed to bind CSK but also failed to phosphorylate CSK efficiently (Fig. 2a, top two panels).

To further examine the interaction between the ACK1 kinase domain and the CSK SH3 domain, we performed docking studies with the structures of full-length CSK (pdb-entry 1K9A)[32] and the ACK1 kinase domain (pdb-entry 4HZR)[33] using ClusPro[34]. ACK1 was used as

the receptor, and CSK was used as the ligand. The restraints were generated for docking based on the cocrystal structure of the insulin receptor kinase (IRK) with a peptide (PDB-entry 1IR3)[35] by aligning IRK on ACK1 and calculating seven Cα-Cα distances between structurally conserved residues in ACK1 and the position of the substrate tyrosine bound to IRK. Multiple parallel docking runs were performed while relaxing the restraints from 2.5 Å to 5 Å. The solution with the most similar backbone geometry around Y18 to that of the substrate peptide observed in the PDB entry 1IR3 was chosen. The structure of IRK (PDB ID: 1IR3) was aligned to ACK1 kinase when complexed with CSK to

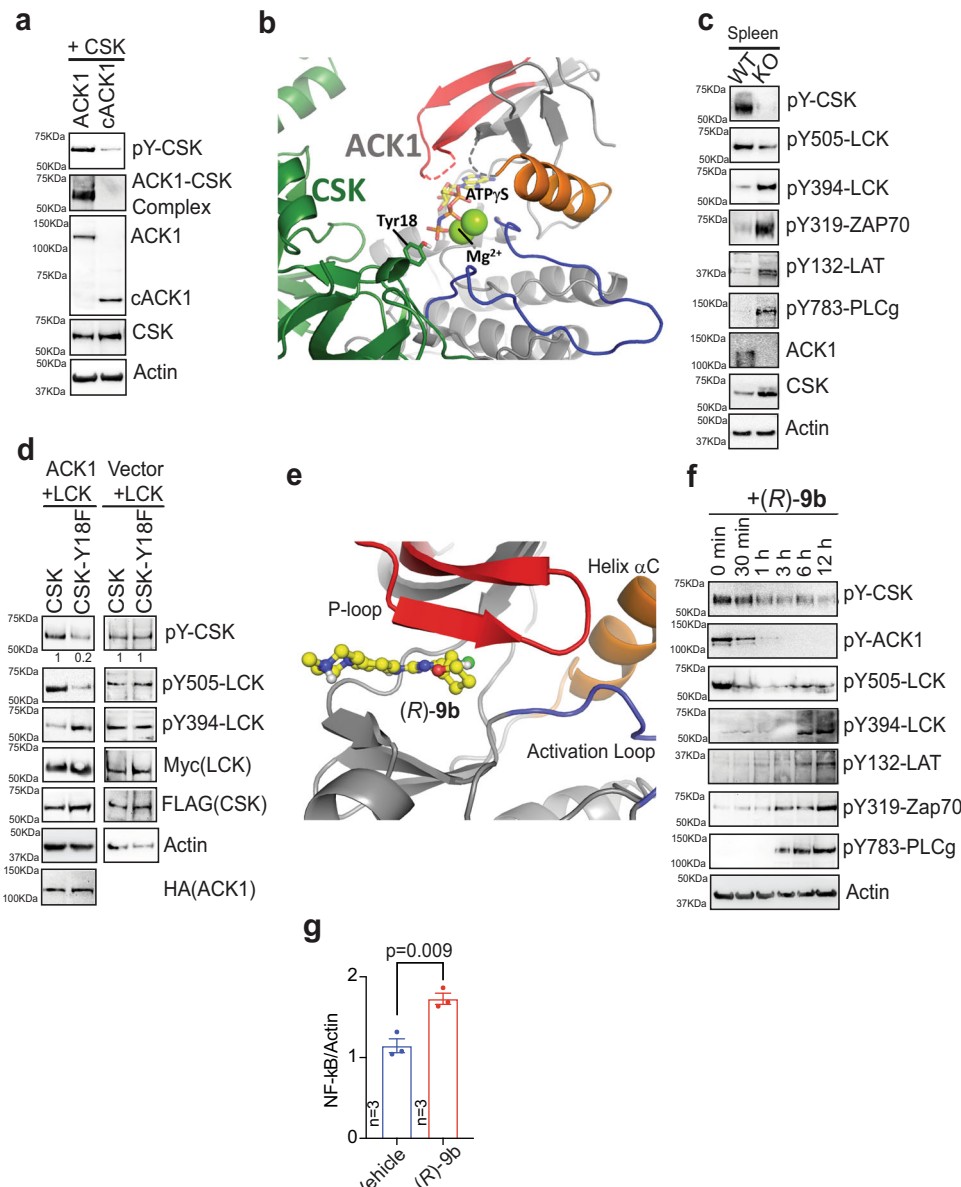

**Fig. 2 | ACK1 phosphorylates CSK at Tyr18 in SH3 domain dampening T cell responsiveness. a** HEK293T cells were co-transfected with FLAG-tagged CSK and HA-tagged ACK1 or cACK1. Post-MG132 (25 μM, 3 hr) treatment, lysates were immunoprecipitated with FLAG beads, followed by immunoblotting with p-Tyr antibodies (top panel). In addition, lysates were immunoprecipitated with HA beads, followed by immunoblotting with FLAG antibodies (2nd panel). **b** Docking model of CSK (green) bound to ACK1 (gray) with ATPγS (yellow)/Mg²⁺ (green) modelled into the active site of ACK1. The SH3 domain of CSK docks against the activation loop of ACK1 (red). **c** Lysates prepared from spleen of WT and *Ack1* KO mice were immunoblotted with the indicated antibodies. Lysates were also immunoprecipitated with CSK antibodies, followed by immunoblotting with p-Tyr antibodies (2nd panel). **d** HEK293T cells were co-transfected with or without ACK1,

LCK and CSK or CSK-Y18F mutant expressing constructs. Lysates were immunoblotted with indicated antibodies. **e** X-ray co-crystal structure of ACK1 kinase domain bound to (*R*)-**9b**. The compound (*R*)-**9b** binds to the ATP binding site underneath the phosphate binding P-loop (red), without interactions to helix αC (orange) or the activation loop (blue). **f** Jurkat cells were treated with (*R*)-**9b** (3 μM) for indicated time periods and the lysates were immunoblotted with indicated antibodies. **g** RNA was prepared from Jurkat cells treated with vehicle or (*R*)-**9b**, followed by real time PCR of NF- κB (*n* = 3 biologically independent samples, 3 replicates). For **a**, **c**, **d** and **f**, representative images are as shown (*n* = 3 biologically independent experiments). For **g**, the data are represented as mean ± SEM and *p* values were determined by unpaired two-tailed Student's *t*-test. Source data are provided as a Source Data file.

position ATP and two Mg2+ ions. The complex was analysed in COOT[36], and the side chain chi-angles of Tyr18 were changed to rotate the side chain hydroxyl group towards ATPγS. The interface between the two protein chains buried 979 Å² with a predicted free energy gain of solvation of −5.5 kcal/mol (PISA server). The complex was stabilized by nine hydrogen bonds and three salt bridges. The primary interactions were identified between the SH3 domain of CSK and the C-lobe of ACK1 around helix αG (residues 242–252). Y18 was only partially solvent-exposed in the isolated structure of CSK and formed an

intramolecular hydrogen bond with Tyr184. Upon adjustment of the Tyr18 side chain chi angles, the hydroxyl group was observed to face the gamma-phosphate of ATP, providing insight into the mechanism of CSK Tyr18 phosphorylation by ACK1 (Fig. 2b). These data suggest that ACK1-mediated CSK Tyr18 phosphorylation is a potential mechanism that maintains T cells in an inactive state in the absence of TCR/CD3-mediated activation.

To validate the role played by ACK1 in CSK Y18 phosphorylation and establish the resulting effects, we generated antibodies against

pY18-CSK (Supplementary Fig. 6a–c). Briefly, dot blot analysis revealed that these antibodies recognized the CSK peptide with the Y18-phosphorylated residue at the centre (pY18-CSK phospho-peptide) but did not recognize a CSK peptide in which Tyr18 was not phosphorylated or one containing an unrelated phospho-peptide (pY37-H2B)[37] (Supplementary Fig. 6a). When these antibodies were incubated with a pY18-CSK phospho-peptide and then performed immunoblotting, the antibodies were found to be neutralized, and no signal was observed (Supplementary Fig. 6b, middle panel). In contrast, when these antibodies were incubated with an unrelated phospho-peptide (derived from histone H2B, pY37-H2B), a signal was observed (Supplementary Fig. 6b, top panel). In addition, antibodies against pY18-CSK were assessed against multiple unrelated phospho-peptides, e.g., peptides derived from phospho-AKT, phospho-ATP synthase, phospho-histones H2A, H2B, H3 and H4. The anti-pY18-CSK antibodies did not cross-react with any of these phospho-peptides (Supplementary Fig. 6c). Furthermore, treatment of cells with the phosphatase inhibitor cocktail significantly increased pY18-CSK levels (Supplementary Fig. 6d, top panel).

Lysates prepared from the spleens of WT and *Ack1*-KO mice were subjected to immunoblotting with the indicated antibodies. A significant reduction in CSK Y18 phosphorylation was observed, resulting in decreased LCK Y505 and increased Y394 phosphorylation in the KO mice, leading to increased phosphorylation of Zap70-Y319, LAT-Y132, and PLCγ-Y783 (Fig. 2c). Furthermore, to determine whether ACK1-mediated Y18 phosphorylation of CSK is primarily involved in LCK activity regulation, the Y18 site was mutated to generate a CSK-Y18F mutant. Coexpression of Myc-tagged LCK and FLAG-tagged CSK or mutant CSK-Y18F with or without HA-tagged ACK1 was performed, and then, immunoprecipitation of LCK and CSK with Myc and FLAG antibodies, respectively, was performed. CSK-Y18F mutant-overexpressing cells exhibited a significant reduction in LCK Y505 phosphorylation and an increase in Y394 phosphorylation (Fig. 2d), suggesting that ACK1-mediated CSK Y18 phosphorylation was required for LCK activity regulation. Together, these data suggest a two-step interaction process wherein the proline-rich regions of ACK1 bind to the CSK SH3 domain, bringing them in proximity such that the ACK1 kinase domain contacts and phosphorylates CSK at Tyr18 in the SH3 domain. Furthermore, the loss of ACK1 kinase activity compromised CSK Y18 phosphorylation, which in turn activated LCK, promoting the phosphorylation of Zap70, LAT, and PLC-γ.

## LCK activation is reinvigorated by the ACK1 kinase inhibitor (*R*)−9b

Next, we sought to determine the effect of in vivo inhibition of ACK1 on CSK-LCK signalling by using a new first-in-class ACK1 small-molecule inhibitor, (*R*)-9b, developed by our group[13]. (*R*)-9b is a potent ACK1 inhibitor that shows high specificity in vitro and in vivo[13,38]. To further examine the details of ACK1 interactions with the inhibitor (*R*)-9b, we expressed and purified the kinase catalytic domain of ACK1 in Sf9 insect cells using a recombinant baculovirus[39]. We resolved the X-ray crystal structure of the ACK1 kinase domain in complex with (*R*)-9b at a 1.79-Å resolution (Fig. 2e, Supplementary Fig. 6e, f and Supplementary Table 1; see also the validation report). The PDB ID for this structure is 7KP6. ACK1 crystallized with two copies of the kinase domain per asymmetric unit. Both copies of the kinase domain show density for the inhibitor and their structures are virtually identical. We therefore discuss only the complex with (*R*)-9b in chain A. The kinase domain is in the active conformation with helix αC rotated towards the active site, and a salt bridge between E177 and the catalytic lysine (K158) was formed. The activation loop was folded underneath helix αC in a conformation capable of substrate peptide binding similar to that observed for active Src and IRK[35,40]. The aspartate of the Asp-Phe-

Gly (DFG) motif was deeply embedded into the active site and formed a hydrogen bond with the side chain of S136 at the turn of the phosphate-binding loop. (*R*)-9b bound to the ATP-binding site underneath the phosphate-binding loop and formed two hydrogen bonds with the backbone of A208 in the hinge region of the kinase domain. The phenyl of (*R*)-9b engaged in hydrophobic interactions with the side chain of L132, and pyrimidine bound to a hydrophobic pocket formed by the side chains of V140, A156, I190, and L259 (Fig. 2e, Supplementary Fig. 6e, f). Since the kinase domain was in the active conformation and (*R*)-9b largely interacted with the hinge region, it was concluded that the binding mode was consistent with that of a type-1 kinase inhibitor.

To assess the outcome of inhibiting ACK1 kinase activity, Jurkat cells, immortalized human T lymphocytes, were treated with (*R*)-9b, and then, immunoblotting was performed with these cells. Within 1–3 h of (*R*)-9b treatment, a significant decrease in ACK1 Y284-, CSK Y18-, and LCK Y505 phosphorylation was observed, which was followed by increased LCK Y394, Zap Y319, LAT Y132, and PLCγ Y783 phosphorylation (Fig. 2f). Flow cytometry analysis showed a significant reduction in CSK Y18 phosphorylation upon (*R*)-9b and anti-CD3 antibody treatment (Supplementary Fig. 6g). Splenocytes isolated from *Ack1*-KO mice also showed a significant reduction in CSK Y18 phosphorylation compared to that in WT mice (Supplementary Fig. 6h). As NF-κB plays a crucial role in T-cell activation, quantitative RT–PCR was performed with Jurkat cells treated with (*R*)-9b. The results showed an increase in the NF-kB level in cells after (*R*)-9b treatment compared to that after vehicle treatment (Fig. 2g). Overall, these data indicate that inhibition of ACK1 kinase activity can relieve the immune-inhibitory activity of CSK.

## Y18 phosphorylation modulates CSK levels by regulating phosphorylation-dependent ubiquitination and degradation

Unexpectedly, the spleen of *Ack1*-KO mice exhibited a significant increase in global CSK levels (Fig. 2c, Panel 9). To determine whether ACK1 kinase activity plays a role in maintaining CSK protein levels, CSK was coexpressed with ACK1 and a kinase-dead mutant (K158R) of ACK1 (kd-ACK1)[41]. kd-ACK1 not only failed to phosphorylate CSK and form a complex with it but also protected the CSK protein from degradation (Fig. 3a, top three panels). To examine whether CSK Y18 phosphorylation causes polyubiquitination and degradation, Jurkat cells were treated with the proteasome inhibitor MG132 for 3 h. Cell lysates were then prepared and subjected to immunoprecipitation with anit-pY18-CSK antibodies, followed by immunoblotting with anti-ubiquitin antibodies. MG132 treatment increased the poly-ubiquitination of endogenous CSK, and this effect was significantly compromised by (*R*)-9b treatment (Fig. 3b, compare Lanes 2 and 4). Furthermore, to determine whether ACK1 kinase activity is needed for CSK polyubiquitination, cells expressing ACK1 or kd-ACK1 were treated with MG132. ACK1 caused the polyubiquitination of CSK; however, kd-ACK1 prevented CSK polyubiquitination (Fig. 3c). Moreover, in contrast to CSK, the CSK-Y18F mutant exhibited a significant reduction in polyubiquitination when coexpressed with ACK1 and treated with MG132 (Fig. 3d).

To explore the temporal regulation of CSK levels, HEK293T cells were cotransfected with ACK1, CSK or CSK-Y18F mutant and cultured in cycloheximide for 0, 2, 4 and 6 hrs. CSK was almost completely lost within 6 h; however, the CSK-Y18F mutant was not degraded (Fig. 3e). The results of pharmacological inhibition of ACK1 were comparable with those after ACK1-knockdown by siRNA in Jurkat cells. A significant reduction ACK1 Y284-, CSK Y18-, and LCK Y505 phosphorylation was observed; in addition, increased LCK Y394- phosphorylation was observed (Fig. 3f). Collectively, these data indicate that ACK1 not only regulates CSK activation but also promotes its proteasomal turnover to prevent excessive CSK activity, possibly allowing LCK to be rapidly activated upon ligand binding.

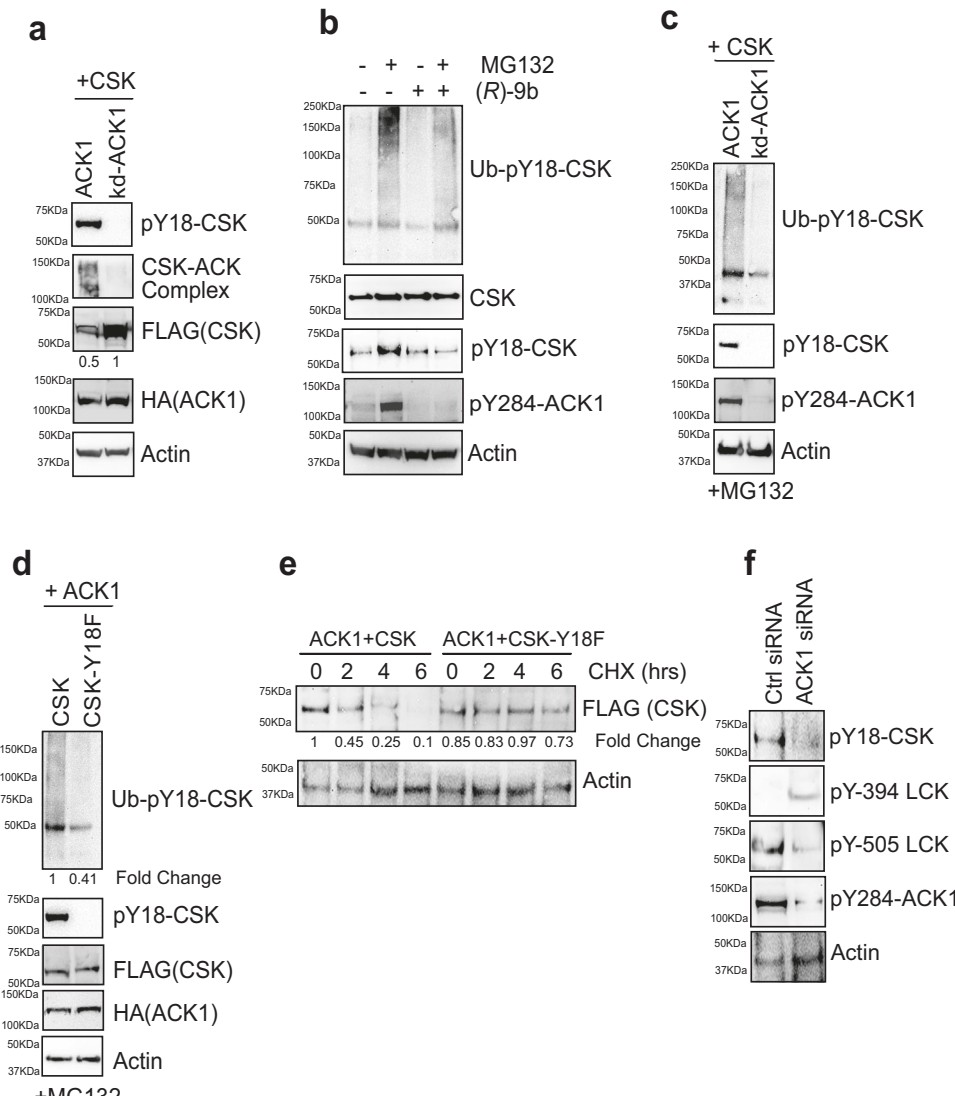

**Fig. 3 | ACK1 mediated CSK Y18-phosphorylation regulates its proteasomal turnover. a** HEK293T cells were co-transfected with FLAG-tagged CSK and HA-tagged ACK1 or kd-ACK1. Lysates were immunoprecipitated with CSK antibody, followed by immunoblotting with ACK1 antibodies (2nd panel). Upper and lower blots were immunoblotted with indicated antibodies. Fold changes were obtained for indicated immunoblots after normalization with respective actin levels. **b** Jurkat cells were treated with DMSO or (R)-**9b** (3 μM) overnight, followed by treatment with MG132 (25 μM, 3 hr). Lysates were immunoprecipitated with pY18-CSK antibody, followed by immunoblotting with ubiquitin antibody (top panel). **c** HEK293T cells were co-transfected with CSK and ACK1 or kd-ACK1. 48 hr post transfection, cells were treated with MG132 (25 μM, 3 hr). Lysates were immunoprecipitated with pY18-CSK antibody, followed by immunoblotting with ubiquitin

antibody (top panel). **d** HEK293T cells were co-transfected with ACK1, CSK or CSK(Y18F) mutant. 48 hr post transfection, cells were treated with MG132 (25 μM, 3 hr) and the lysates were immunoprecipitated with pY18-CSK antibody, followed by immunoblotting with ubiquitin antibody (top panel). **e** HEK293T cells were co-transfected with ACK1, CSK or CSK(Y18F) mutant. Post 48 hr of transfection, cells were cultured in 1% serum and Cycloheximide (10 μM) for 0, 2, 4 and 6 hrs. Thereafter, lysates were immunoblotting with pY18-CSK and Actin antibody. **f** Jurkat cells were transfected with control (Ctrl) or ACK1 siRNA. 48 hr post transfection, the lysates were immunoblotting with indicated antibodies. Representative images are as shown (n = 3 biologically independent experiments). Source data are provided as a Source Data file.

## Genetic ablation of *Ack1* impairs syngeneic tumour growth

To explore the outcome of the immunoinhibitory activity of ACK1-CSK signalling, we analysed a TRAMP-C2 syngeneic mouse model system[42,43]. TRAMP-C2 cells ($1.2 \times 10^6$ cells) were injected subcutaneously (s.c.) into male WT and *Ack1* KO mice. Tumour growth measurement was initiated 4 weeks postinjection, and approximately 11 weeks postinjection, the mice were euthanized, and the tumours were excised. WT mice exhibited tumour growth, reaching an average tumour volume of ~1500 mm³; however, the *Ack1* KO mice exhibited a marked reduction in tumour growth (Fig. 4a and Supplementary Fig. 7a). Analysis of the harvested tumours revealed that although all 10 WT mice formed tumours, only 3 of 10 *Ack1* KO mice developed tumours, and no trace of tumour, either

subcutaneous or in any organ, was detected in the 7 remaining *Ack1* KO mice (Supplementary Fig. 7b). To elucidate the reason for this tumour suppressive effect, mouse spleens were harvested, and subjected to immunophenotyping, which revealed a significant increase in the number of effector CD44^hiCD62L^low CD8⁺ T cells, CD137⁺/CD8⁺ cytotoxic T cells and CD69⁺/CD4⁺ cells in *Ack1* KO mice compared with WT mice (Fig. 4b–d and Supplementary Fig. 7c–e). In addition, draining lymph nodes revealed a significant increase in the number of CD4⁺ and CD8⁺ cells in *Ack1* KO mice (Fig. 4e and Supplementary Fig. 7f). Furthermore, RNA was isolated from the splenocytes, and the expression of IL-2 and IFN-γ was assessed; their levels were significantly increased in the tumour-implanted *Ack1* KO mice (Fig. 4f, g).

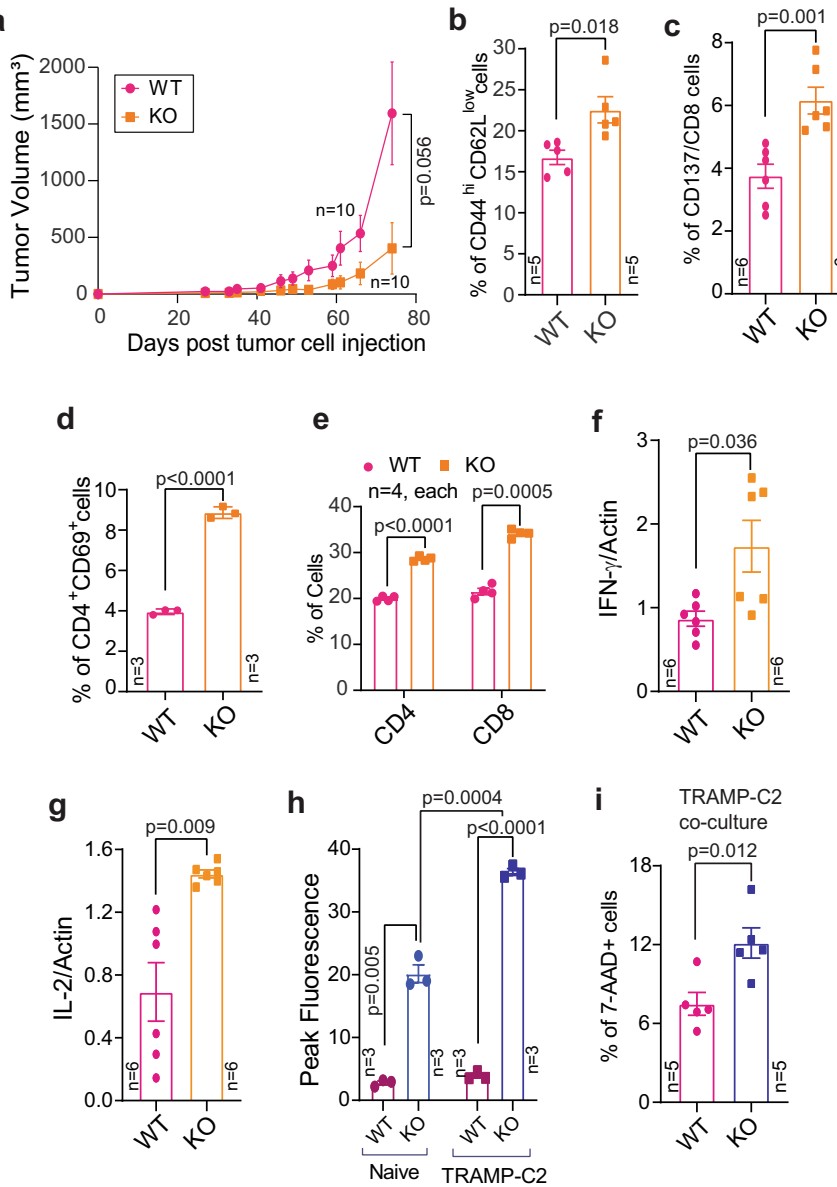

**Fig. 4 | ACK1 negatively regulates the TCR activation diminishing T cell response. a** TRAMP-C2 cells were implanted subcutaneously in 6- to 8-week-old WT and *Ack1* KO male mice and tumor volumes were measured with calipers (*n* = 10 mice in each group). Splenocytes were isolated from WT and *Ack1* KO mice implanted with TRAMP-C2 cells. Flow cytometry analysis was performed to assess CD44hiCD62Llow on CD8 gated population (**b**) (*n* = 5 mice in each group); CD137 on CD8 gated population (**c**) (*n* = 6 mice in each group); and CD69 expression on CD4 gated population (**d**) (*n* = 3 mice in each group). **e** Lymphocytes were drained from lymph nodes of WT and *Ack1* KO mice implanted with TRAMP-C2 cells. Flow cytometry analysis was performed to assess CD4 and CD8 populations (*n* = 4 mice in each group). RNA isolated from the splenocytes was subjected to qRT-PCR with IFN-γ (**f**), IL-2 (**g**) and actin primers (*n* = 6 biologically independent samples, three

replicates). For **a–g**, the data are represented as mean ± SEM from two independent experiments. **h** Splenocytes were incubated with TRAMP-C2 cells and calcium flux was determined using dye Fluo-8; the representative peak fluorescence counts corresponding to calcium flux from three independent mice experiments are shown (*n* = 3 mice in each group). **i** WT and *Ack1* KO mice splenocytes were co-cultured with TRAMP-C2 cells. Tumor cell death was assessed by flow cytometry using 7-AAD staining. Percentage of positive cells are shown (*n* = 5 mice in each group). For **a**, *p* value was determined by paired two-tailed Student's *t*-test. For **b–g**, *p* value was determined by unpaired two-tailed Student's *t*-test. For **h** and **i**, the data are represented as mean ± SEM from three independent experiments and *p* value was determined by unpaired two-tailed Student's *t*-test. Source data are provided as a Source Data file.

To assess whether the splenocytes of *Ack1*-KO mice exhibit a superior T-cell response compared to that of WT mice when challenged with prostate tumour antigen, we performed calcium flux measurements. Splenocytes isolated from WT and *Ack1* KO mice were incubated with TRAMP-C2 cells; *Ack1* KO mice splenocytes exhibited significantly higher calcium responses than those of WT mice (Fig. 4h). We further examined whether the loss of ACK1 leading to T-cell activation is the major cause of tumour suppression. A cell-killing assay was performed; splenocytes isolated from WT and *Ack1* KO mice were

cocultured with TRAMP-C2 cells. Cytotoxic T-cell-mediated cancer cell killing was assessed by staining with 7-AAD, and 7-AAD+ cells were quantified via flow cytometry. A significant increase in TRAMP-C2 cell death was observed upon incubation with KO splenocytes (Fig. 4i).

To corroborate the observed T-cell activation in the spleen and tumour growth inhibition, *Ack1* KO and WT mice were injected with TRAMP-C2 cells, and the tumour infiltrating lymphocytes (TILs) were analyzed. Tumours were digested with collagenase, and single-cell suspensions were stained with a live/dead discriminator and various

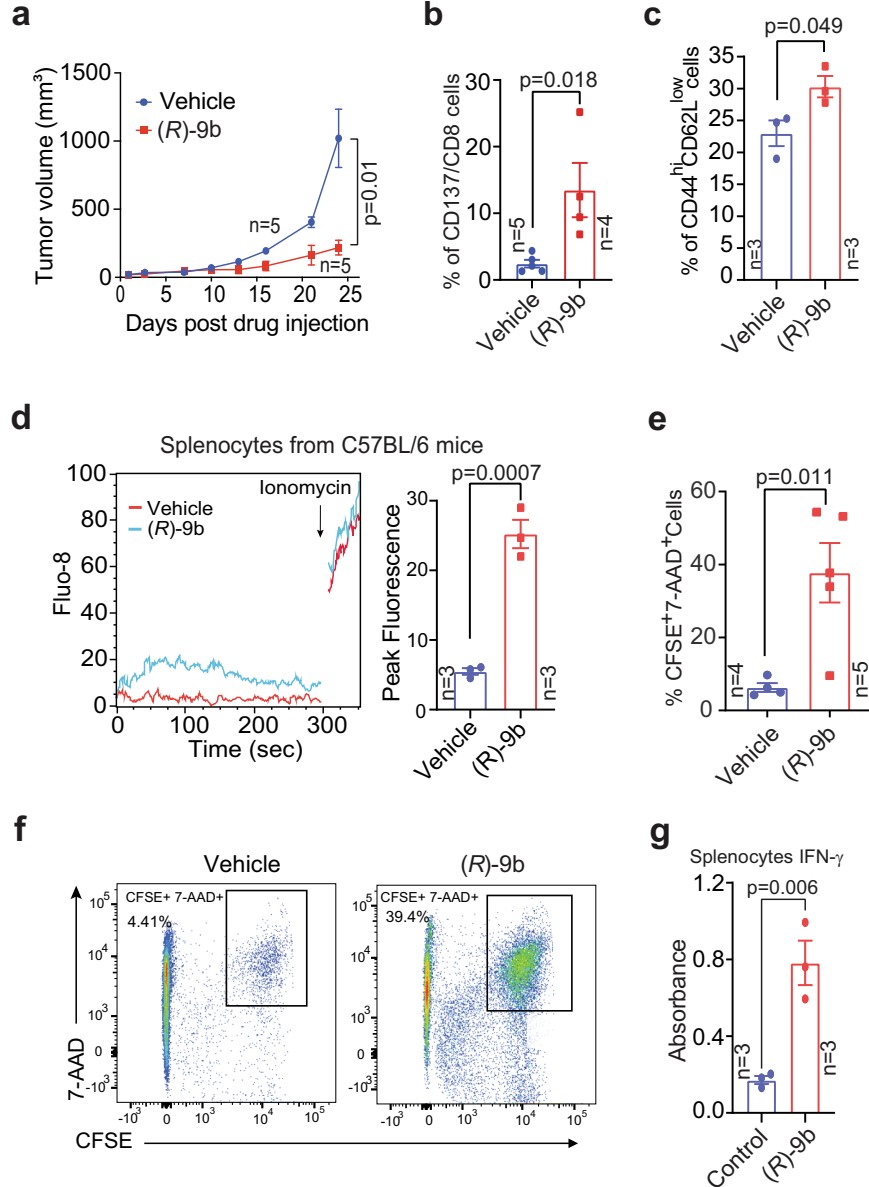

**Fig. 5 | Inhibition of ACK1 kinase activity augments the TCR response threshold against prostate tumors. a** TRAMP-C2 cells implanted 6- to 8-week-old C57BL/6 male mice were injected either with vehicle (6% Captisol) or (*R*)-**9b** in 6% Captisol. Tumor volumes were measured with calipers (*n* = 5 mice in each group. Data are represented as mean ± SEM, and *p* value was determined by paired two-tailed Student's *t*-test. **b**, **c** Splenocytes were isolated from vehicle and (*R*)-**9b** injected mice implanted with TRAMP-C2 cells. Flow cytometric analysis was performed to assess CD137 on CD8 gated population (**b**), and CD44$^{hi}$CD62L$^{low}$ on CD8 gated population (**c**). **d** Representative calcium flux and the corresponding peak fluor-escence counts of splenocytes from C57BL/6 male mice described above. The peak

fluorescence is shown on right (*n* = 3 mice in each group). **e**, **f** Treated splenocytes were co-cultured with CFSE stained TRAMP-C2 cells. Tumor cell lysis was assessed by flow cytometry using 7-AAD staining. Percentage of double positive cells are shown. Representative scatter plots are shown (**f**). **g** Splenocytes isolated from C57BL/6 mice were treated overnight with (*R*)-**9b** and levels of IFN-γ in the culture supernatant was determined by ELISA (*n* = 3 mice in each group). For **b**–**d**, **e** and **g**, the data are represented as mean ± SEM and *p* values were determined by unpaired two-tailed Student's *t*-test. Source data are provided as a Source Data file.

antibodies. The first gate excluded nonlymphocyte populations based on forward and side scatter (FSC and SSC). Following this, CD45 gating was performed to identify total lymphocytes, and CD3$^+$ cells were then gated. The CD3 by CD4/CD8 gate allowed us to identify CD8$^+$ or CD4$^+$ TILs. A significant increase in CD4$^+$ CD69$^+$ cells expressing perforin was observed in KO mice (Supplementary Fig. 8a). In contrast, a decrease in the number of Tregs was observed in KO mice (Supplementary Fig. 8b). Additionally, there was a decrease in the levels of the cell exhaustion markers, namely PD1, Tim3 and Lag3 in the TILs in *Ack1* KO mice compared to WT mice (Supplementary Fig. 8c). Similar to the TRAMP-C2 tumours, WT and *Ack1* KO mice were challenged with B16

melanoma cells to create a tumour burden. Consistent with the data shown in Fig. 4a, a significant decrease in the volume of B16 tumours was observed (Supplementary Fig. 8d). Overall, these data indicate that the loss of ACK1 lowers the threshold for T cell activation, augmenting their activity when exposed to prostate tumour antigen.

## Pharmacological inhibition of ACK1 impedes syngeneic tumour growth

A major barrier for immunotherapeutic approaches in prostate cancer is marked immunosuppression within the tumour milieu. To explore whether ACK1-mediated CSK Y18 phosphorylation is a

signalling event driving immunosuppression, we aimed to elucidate the effect of ACK1 inhibition on CRPC tumour growth. TRAMP-C2 cells, which are castrate-resistant (or CRPC type), were implanted in C57BL/6 mice, and 5 weeks post-injection, when tumours were palpable, the mice were subcutaneously injected with (R)-9b at 24 mg/kg five times per week, for approximately 4 weeks. In another experiment, (R)-9b was injected intratumorally five times per week for approximately 3.5 weeks. The mice injected with (R)-9b exhibited a marked decrease in tumour growth (Fig. 5a, Supplementary Fig. 9a, b). Harvested spleens from the (R)-9b-treated mice revealed a significant increase in the number of CD137+/CD8+ cytotoxic T cells, effector CD44hiCD62Llow CD8+ T cells (Fig. 5b, c, and Supplementary Fig. 9c, d) and CD4+ CD69+ cells expressing perforin (Supplementary Fig. 10a) compared with those in vehicle-injected mice. Tumour draining lymph nodes also showed an increase in the number of effector CD44hiCD62Llow CD8+ T cells in the (R)-9b-injected mice (Supplementary Fig. 10b). T-cell activation was also reflected in increased calcium flux (Fig. 5d).

To verify that inhibition of the pY-ACK1/pY18-CSK/pY505-LCK signalling circuit by (R)-9b is primarily responsible for suppressing tumour growth, splenocytes from C57BL/6 mice were isolated and treated with (R)-9b. After the compound was washed off, a cell-killing assay was performed wherein CFSE-stained TRAMP-C2 cells were incubated with splenocytes. A significant increase in TRAMP-C2 cell death (CFSE+ 7-AAD+ cells) was observed for the splenocytes treated with (R)-9b compared with those treated with vehicle (Fig. 5e, f). Furthermore, ELISA of the culture supernatant revealed a significant increase in IFN-γ levels upon (R)-9b treatment (Fig. 5g). A flow cytometry analysis of splenocytes confirmed the increased levels of IL-2 and IFN-γ expression in the T cells in (R)-9b-treated mice (Supplementary Fig. 10c, d). There was a decrease in the level of cell exhaustion markers (PD-1, Lag3, and Tim3) on CD8+ T cells (Supplementary Fig. 11a, b). The assessment of the CD8 T-cell memory pool after increasing the tumour burden showed an increase in the number of central memory and effector memory T cells upon ACK1 loss induced either by knockdown or pharmacological inhibition (Supplementary Fig. 12a, b), indicating that ACK1 inhibition leads to higher T-cell responses, inducing for long-term memory for tumour antigens.

### Loss of Ack1 impairs tumour growth by activating T cells and promoting persistent immune surveillance

To validate that TRAMP-C2 tumour growth is compromised in Ack1 KO mice because of increased T-cell activation, we assessed the effect of antibody-mediated depletion of T cells. Starting 3 days prior to TRAMP-C2 cell implantation, WT and Ack1 KO mice were injected intraperitoneally with anti-CD4, anti-CD8 or both types of depletion antibodies once per week for 5 weeks. Consistent with Fig. 4a, WT mice injected with IgG exhibited tumour growth, while Ack1 KO mice injected with IgG exhibited negligible tumour growth. However, Ack1 KO mice injected with anti-CD4 and anit-CD8 or both types of depletion antibodies significantly inhibited tumour growth (Fig. 6a, Supplementary Fig. 13a, b). The prostate tumour microenvironment in both mice (prostate tumours derived from TRAMP mice) and humans expressed MHC class I and II[44]. Analysis of CAR-T cells that can persist for ten years in CLL patients revealed a predominant cytotoxic CD4+ T-cell population, raising the possibility that CD4+ T cells play important roles in durable T-cell therapy[45]. Taken together, these data suggest that CD4+ T cells exhibit cytotoxic characteristics with functional activation, which can play a role in controlling tumour growth.

Furthermore, to corroborate the role of ACK1 kinase in regulating T-cell activation, adoptive transfer of purified T cells from WT and Ack1 KO mice into TRAMP-C2-implanted NSG mice was performed. Purified T cells ($1.5 \times 10^6$) were injected once per week from the 4th day post-TRAMP implantation. The mice injected with T cells from Ack1 KO mice showed significant tumour growth regression compared to that in the mice injected with WT T cells (Fig. 6b and Supplementary Fig. 13c). Isolation of splenocytes followed by flow cytometry revealed an increase in the percentage of effector CD8+ T cells in the mice injected with T cells from Ack1 KO mice compared to the mice injected with WT T cells (Fig. 6c). Analysis of TILs from the tumours of these NSG mice showed a significant increase in infiltrated lymphocytes in the tumours of mice that had been injected with T cells from the Ack1 KO mice (Supplementary Fig. 13d). Furthermore, the phenotype of the TILs revealed an increased number with CD44hiCD62Llow expression (Fig. 6d and Supplementary Fig. 13e).

One of the primary functions of immune surveillance during carcinogenesis is the development of antigen-specific T cells that can potentiate appropriate inhibitory effector functions. SPAS-1 (stimulator of prostatic adenocarcinoma-specific T cells) is a prostate tumour antigen that is a major T-cell target in TRAMP tumours[46]. To confirm that prostate tumour-specific T cells suppressed tumour growth, TRAMP tumour-bearing NSG mice were injected with T cells from Ack1 KO or WT mice (Fig. 6e). The T cells were isolated from the spleen and the draining lymph nodes and stained with SPAS-1 tetramer or control tetramer. Flow cytometry revealed an increased number of prostate cancer-specific T cells (CD45+ CD3+ SPAS-1+) in Ack1 KO mice compared to WT mice (Fig. 6f, g). The control tetramer data are shown in Supplementary Fig. 13f.

The persistence of the T cells from Ack1 KO and WT mice was assessed by adoptively transferring CFSE-stained T cells into TRAMP-C2 tumour-bearing NSG mice. Periodic submandibular blood collection and flow cytometry of lymphocytes showed that the T cells from the KO mice showed better homing and higher proliferation than the T cells from the WT mice (Supplementary Fig. 14a, b). Collectively, these data indicate that the inhibition of ACK1 kinase activity augments T-cell responsiveness, imparting sustained immune surveillance against prostate tumours.

### ACK1/EZH2/H3K27me3 epigenetic signalling regulates CXCL10 expression in prostate tumours

Most solid cancers, including prostate cancer, are noninflamed or immune desert tumours where little or no immune cell infiltration is observed, suggesting that T-cell priming or T-cell trafficking to the tumour has failed. We observed that inhibition of ACK1 by (R)-9b promoted persistent immune surveillance, impeding tumour growth in syngeneic models of prostate cancer, suggesting that, in addition to T-cell activation, the tumour microenvironment is also involved in tumour growth. To explore this possibility further, we performed RNA sequencing of CRPC cells (C4-2B), followed by assessment of global cytokine expression profile, which revealed a significant increase in CXCL10 after (R)-9b treatment (GEO accession GSE211835). T-cell migration into the tumour microenvironment depends on the local production of specific chemokines, especially CXCL9 and CXCL10, which engage CXCR3 on the surface of CD8+ effector T cells[47–51]. These chemokines are produced by tumour cells and are also secreted by stromal cells in the tumour microenvironment and thereby contribute to the recruitment of effector T cells[47,51]. We reasoned that CXCL10 transcriptional repression can be realized through the deposition of repressive epigenetic marks, facilitated by ACK1. Histone methyltransferases (HMTases), EZH2 deposits H3K27 trimethylation epigenetic marks, causing transcriptional repression[52], including repression of CXCL10 transcription, in hepatic tumour cells[53]. We performed coimmunoprecipitation studies and found that a kinase activity-dependent interaction of ACK1 with EZH2 (Fig. 7a). Upon inhibition of ACK1, the EZH2 levels were significantly reduced, suggesting EZH2 degradation when not in complex with ACK1 (Fig. 7a, middle panel).

Furthermore, chromatin immunoprecipitation (ChIP) was performed with anti-EZH2 and anti-H3K27me3 antibodies, and a

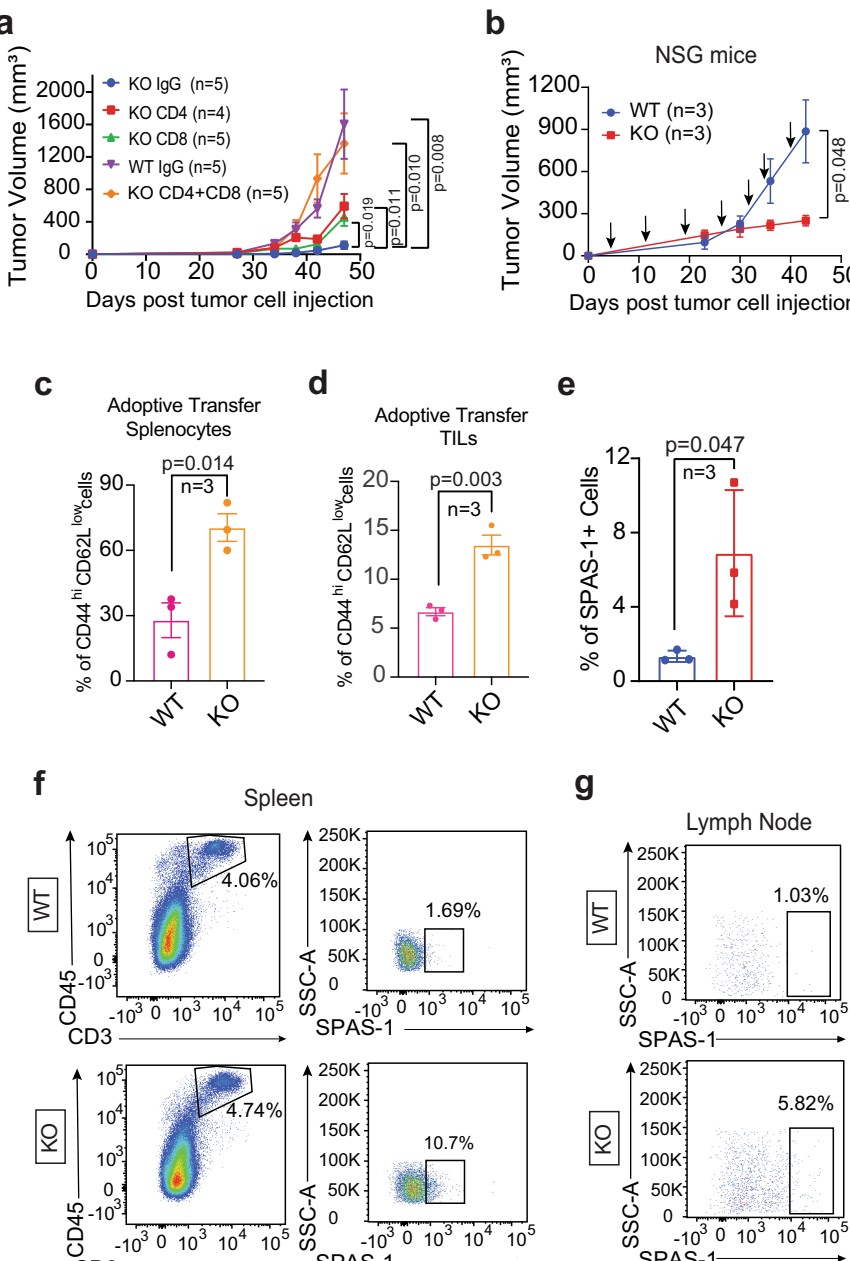

**Fig. 6 | ACK1 knockdown instigates T cell recruitment restraining tumor growth. a** TRAMP-C2 cells were implanted subcutaneously in WT and *Ack1* KO male mice. Anti-CD4 or Anti-CD8 (or IgG as control) depletion antibodies were injected, and tumor volumes were measured with calipers (*n* = 5 in KO IgG, KO CD8, KO CD4+CD8 and WT IgG; *n* = 4 in KO CD4 group). **b**–**g** Adoptive transfer of *Ack1* KO and WT T cells into tumor bearing NSG mice. TRAMP-C2 cells were implanted subcutaneously in NSG mice. Adoptive transfer of purified T cells from WT and *Ack1* KO mice was done once a week starting from the 4th day post-tumor implantation. The tumor volumes were measured with calipers (**b**). After 6 weeks, splenocytes were isolated (**c, e** and **f**), TILs were purified (**d**) and lymph nodes were drained (**g**). The levels of CD44hiCD62Llow on CD8 gated population (**c** and **d**) and SPAS-1 expression (**e**) was assessed by flow cytometry (*n* = 3 mice in each group). Representative scatter plots from three independent experiments are shown (**f** and **g**). For **a**–**e**, the data are represented as mean ± SEM and *p* values were determined by unpaired two-tailed Student's *t*-test. Source data are provided as a Source Data file.

significant decrease in EZH2 recruitment and H3K27me3 deposition at the *CXCL10* promoter was observed after (*R*)-**9b** treatment (Fig. 7b). Then, a significant increase in CXCL10 mRNA expression was observed in C4-2B cells and TRAMP cells that had been treated with (*R*)-**9b** (Fig. 7c). Correspondingly, a significant increase in Cxcr3 mRNA expression was observed in TILs obtained from mice treated with (*R*)-**9b** (Supplementary Fig. 14c). Together, these data indicate that activated ACK1/EZH2 epigenetic signalling in tumour cells depletes the tumour microenvironment of the T-cell attractant CXCL10, maintaining T cells in an inactive state.

## ICB treatment renews pY-ACK1/pY18-CSK/pY505-LCK signalling

T-cell exhaustion, characterized by functional impairment and surface abundance of multiple inhibitory receptors, including PD-1, CTLA-4, TIM-3, and LAG-3, among other markers, has been well established[54–57]. We explored the effect of ICB in prostate cancer, both alone and in combination with (*R*)-**9b**. TRAMP-C2 cell-implanted C57BL/6 mice were treated with ICB (anti-CTLA-4 and anti-PD-1 antibodies; 100 µg of each/mouse), (*R*)−**9b** (20 mg/kg, five times per week, for approximately 4 weeks) or both ICB and (*R*)-**9b**. ICB exhibited a modest effect on tumour growth, which was consistent

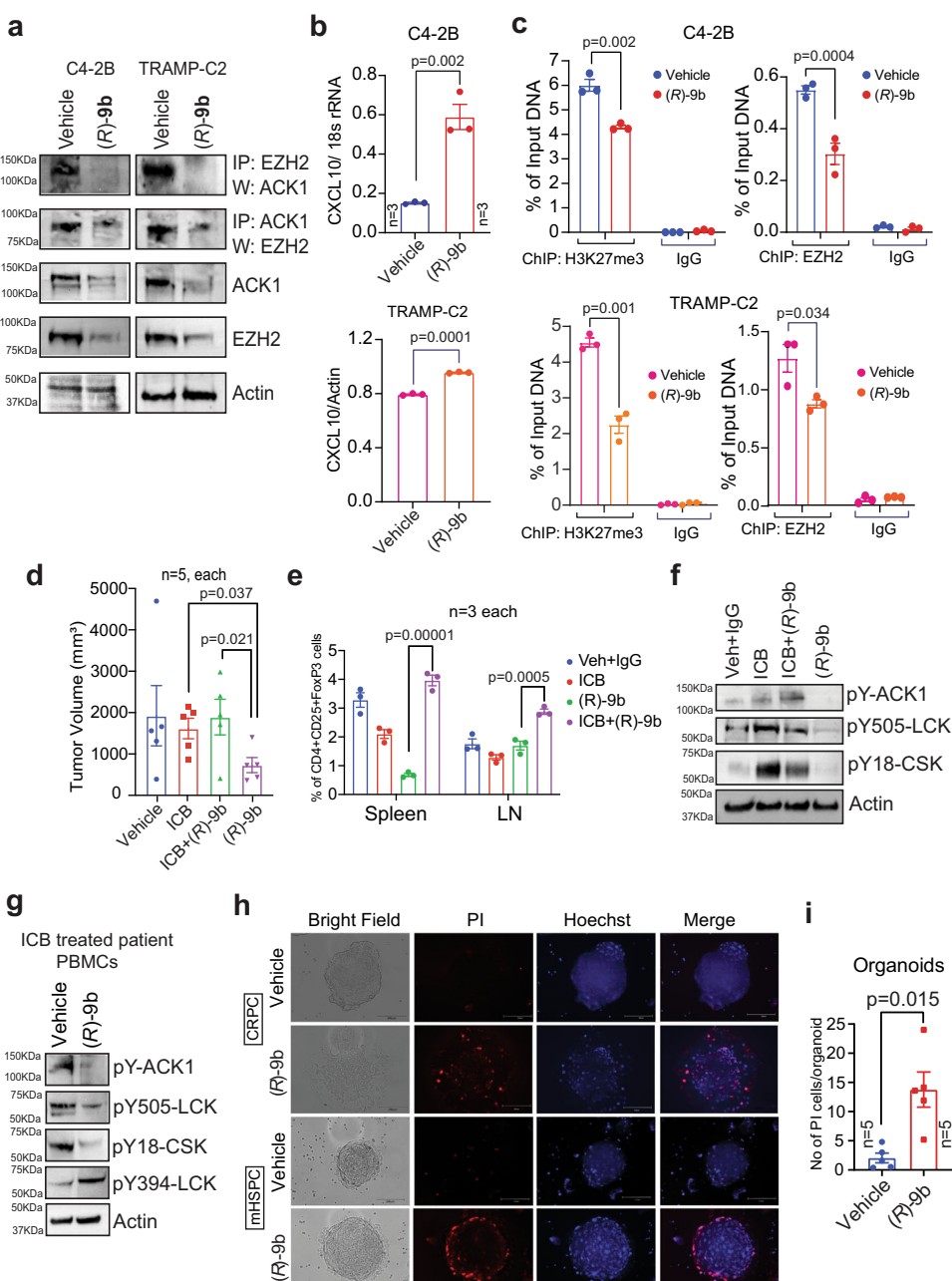

**Fig. 7 | Pharmacological ablation of ACK1 activates CXCL10 in tumors endows human PBMCs with enhanced T cell responsiveness and compromise prostate cancer organoid growth. a** C4-2B and TRAMP-C2 cells were treated with either vehicle (DMSO) or (*R*)-**9b**. Lysates were immunoprecipitated with ACK1 or EZH2 antibody, followed by immunoblotting with indicated antibodies. Representative images are as shown (*n* = 3 biologically independent experiments). **b** C4-2B and TRAMP-C2 cells were treated with either vehicle (DMSO) or (*R*)-**9b**. Total RNA was isolated, followed by qRT-PCR with CXCL10 and actin or 18s rRNA primers (*n* = 3 biologically independent samples, 3 replicates). **c** C4-2B and TRAMP-C2 cells were treated with either vehicle or (*R*)-**9b**, and harvested. The lysates were subjected to ChIP using H3K27me3 and EZH2 antibodies, followed by qPCR with primers for *CXCL10* promoter region (*n* = 3 biologically independent samples, 3 replicates). **d** TRAMP-C2 cells were implanted subcutaneously in 6- to 8-week-old C57BL/6 male mice and were injected with vehicle (6% Captisol)+IgG antibodies, (*R*)-**9b**, αPD-1 and

αCTLA-4 antibodies (ICB), or its combination with (*R*)-**9b** (*n* = 5 mice in each group). **e** Splenocytes were isolated and lymph nodes were drained to assess the Treg (CD4⁺CD25⁺FoxP3⁺) population (*n* = 3 mice in each group). **f** Representative immunoblots of spleen lysates from mice described above are shown (*n* = 3 biologically independent experiments). **g** PBMCs were isolated from blood of ICB treated mHSPC patient. The PBMCs were treated with (*R*)-**9b** for 6 h and immunoblotting was performed with the indicated antibodies. **h** The PBMCs from prostate cancer patients were treated with vehicle or (*R*)-**9b** for 6 h, washed and co-cultured with prostate cancer organoids. After 24 h, the organoids were stained with PI and Hoechst 33258 and observed under fluorescent microscope. Representative images are as shown (*n* = 3 biologically independent experiments). **i** Quantitation of the data shown in **h**. For **b–e** and **i** the data are represented as mean ± SEM and *p* values were determined by unpaired two-tailed Student's *t*-test. Source data are provided as a Source Data file.

with the ICB-refractory nature of CRPCs; in contrast, (*R*)-**9b** treatment significantly suppressed tumour growth. Combining ICB with (*R*)-**9b** did not yield further sensitization of tumour growth, and tumours continued to grow at this concentration of (*R*)-**9b** (Fig. 7d

and Supplementary Fig. 14d). The inability of ICB in combination with (*R*)-**9b** to suppress tumour growth was intriguing, and an analysis of immune cells in the splenocytes and draining lymph nodes showed a significant increase in the Treg population after the ICB + (*R*)-**9b**

treatment compared to that observed after the (*R*)-**9b** alone treatment (Fig. 7e and Supplementary Fig. 15). Together, these data indicate that immune surveillance exerted by (*R*)-**9b** includes suppression of Treg function. Furthermore, immunohistochemistry showed that tumours treated with (*R*)-**9b** as a single agent led to the highest number of infiltrated T cells compared to that induced by the vehicle+IgG-, ICB- or ICB + (*R*)-**9b**-treatment (Supplementary Fig. 16a, b).

Immunoblotting of tumour samples revealed ACK1 activation upon ICB treatment (Fig. 7f). These data revealed an unexpected modality of ICB resistance, namely, reactivation of pY-ACK1/pY18-CSK/pY505-LCK signalling, which further suggested that continued ICB treatment of prostate cancer patients may be counterproductive; however, ACK1 inhibitors are likely to have desirable effects. To validate these suppositions, peripheral blood mononuclear cells (PBMCs) were isolated from the blood of ICB-treated PCa patients. ICB-treated PCa patients exhibited pY-ACK1, pY18-CSK and pY505-LCK expression (Fig. 7g), which was significantly suppressed upon (*R*)-**9b** treatment. Moreover, a significant increase in pY394-LCK levels was observed after (*R*)-**9b** treatment. Taken together, these data suggest that renewed pY-ACK1/pY18-CSK/pY505-LCK signalling in ICB-insensitive prostate cancer could be sensitized by (*R*)-**9b** treatment.

### The ACK1 inhibitor (*R*)-**9b** revives immune-impaired PBMCs from CRPC patients

Lack of tumour infiltration by immune cells is one of the reasons for the ineffectiveness of immune therapies in PC patients. Since (*R*)-**9b** treatment not only activated T cells but also increased CXCL10 expression by cancer cells, we explored collective outcomes using human prostate cancer spheroids that we generated using CRPC-forming C4-2B cells. The PBMCs obtained from prostate cancer patients were treated with (*R*)-**9b**, and after the compound was washed off, the PBMCs were incubated with C4-2B spheroids. Propidium iodide (PI) staining revealed significant spheroid death when incubated with (*R*)-**9b** treated PBMCs. In contrast, spheroids incubated with vehicle-treated PBMCs were unaffected (Supplementary Fig. 16c). Additionally, there was no significant apoptosis of spheroid cells when (*R*)-**9b** was directly added to the spheroids at the concentration used to pretreat the PBMCs in culture (Supplementary Fig. 16d), indicating that a relatively low concentration of (*R*)-**9b** was sufficient to efficiently activate a T-cell responses against tumour antigen.

In addition, we generated human prostate cancer organoids from freshly harvested prostate tumours. These organoids provide a close match to in vivo biology and patient genetics, making them a suitable model system for studying the effects of pharmacological agents. Prostate cancer organoids were incubated with PBMCs purified from metastatic hormone-sensitive prostate cancer (mHSPC) or mCRPC patients and that had been treated with either vehicle or (*R*)-**9b**. As observed with the C4-2B spheroids, significant organoid death was observed upon incubation with PBMCs activated by (*R*)-**9b** treatment (Fig. 7h, i). Immune cell infiltration was particularly noticeable in the outermost layer of the organoids (Fig. 7h).

To further examine the translational relevance of the suppression of ACK1 kinase activity and its role in reversing tumour growth, PBMCs isolated from mCRPC patients were enriched for T cells and treated with (*R*)-**9b**. After the compound was eliminated by washing, PBMCs were cocultured with human CRPCs, C4-2B, or LAPC4 cells. The viability of C4-2B and LAPC4 cells was significantly compromised when cocultured with (*R*)-**9b**-treated CRPC PBMCs compared with vehicle-treated samples (Supplementary Fig. 17a, b). We also assessed the response in healthy donor T cells after (*R*)-**9b** treatment. Immunoblotting showed that (*R*)-9b caused a decrease in pY18-CSK level and a significant increase in pY394-LCK and pY319-Zap70 levels (Supplementary Fig. 17c). These outcomes are reflected in improved calcium responses, as shown in Supplementary Fig. 17d.

## Discussion

The inability of the host immune system to mount a robust T-cell response against malignancies has long perplexed researchers and has led to extensive studies. The initiation of T-cell signalling is dictated by precise Tyr-phosphorylation events executed by a set of specific tyrosine kinases, which are activated in orderly fashion. We discovered a player in the initiation of TCR activation, ACK1, which acts as an upstream tyrosine kinase and inhibits TCR activation by phosphorylating CSK at Tyr18, generating a *rendezvous point* for further interactions (Fig. 8). PAG-Y317 phosphorylation recognition by the SH2 domain of CSK[58,59] brings the CSK SH3 domain in proximity to the ACK1 kinase domain, thus promoting Y18 phosphorylation (Fig. 2b, c). It is likely that ACK1 initiates another contact; for example, the proline-rich region of ACK1 may interact with the SH3 domain of CSK (Fig. 2a), further stabilizing the ACK1-CSK interaction (Fig. 8). In turn, Y18-phosphorylated CSK may inhibit Y505 phosphorylation of LCK, maintaining LCK in an inactive conformation, suggesting that the primary function of ACK1 in immune cells is to prevent ligand-independent TCR activation. Significantly, genetic ablation of *Ack1* or pharmacological inhibition of the ACK1 by (*R*)-**9b** not only prevented CSK Y18- and LCK Y505-phosphorylation but also allowed LCK to regain its kinase activity via Y394- phosphorylation, subsequently causing TCR activation. It is likely that ACK1-mediated CSK Y18 phosphorylation has an additional functional relevance: following LCK Y505 phosphorylation, pY18-CSK is targeted for polyubiquitination and degradation to subdue its disproportionate LCK inhibitory activity, which may prevent efficient Y394 phosphorylation after a ligand becomes available.

PAG/Cbp phosphorylation was not greatly affected in *Ack1* KO mice or Jurkat cells treated with (*R*)-**9b** (Supplementary Fig. 17e, f). PAG/Cbp harbours nine tyrosine residues that can be phosphorylated in T cells by FYN or SRC[58,60]. In addition to Y317, we identified Y227 and Y417 phosphorylation in PAG/Cbp (Supplementary Fig. 5b–d), suggesting that multiple cytoplasmic tyrosine kinases can target PAG/Cbp, including ACK1, and this diversity may compensate for a loss of FYN and SRC activity or confer subtype specificity (or both).

The identification of two distinct signalling pathways operational in parallel in different cells, namely, cancer cells and T cells, driven by the same kinase was the prime observation of this study. The ACK1 kinase not only suppressed T-cell activation but also a key chemokine, CXCL10 (Fig. 7a–c), which plays a crucial role in directing the migratory properties and potentiation of CD4⁺ and CD8⁺ T cells in the context of cancer and inflammatory autoimmunity[61]. CXCL10 acts as a ligand for CXCR3, a chemokine receptor that is abundant on CD4⁺ and CD8⁺ T cells and NK cells[53]. Thus, (*R*)-**9b** perturbs the prostate cancer microenvironment by contributing two important cancer-inhibiting functions; host T-cell activation and increased T-cell navigation to tumour cells, by suppressing phosphorylation of a single tyrosine kinase, ACK1.

Increased immune activation upon loss of ACK1 opens up the possibility of autoimmunity in organs; however, the ALT assay and anti-nuclear antibody (ANA) assay showed little difference and minimal signs of inflammation or fibrosis (Supplementary Fig. 4g, h). However, the possibility that loss-of-function mutations in *ACK1* predispose patients to autoimmune diseases cannot be ruled out.

The dependence of prostate cancer on androgen receptor (AR) signalling has been extensively exploited; for example, second-generation AR antagonists such as enzalutamide and apalutamide or the androgen synthesis inhibitor abiraterone have been widely used[20,62–65]. Recently, the association between the transcriptional coactivator MED1 and AR was discovered as a vulnerability in prostate cancer[66]. Notwithstanding an initial response, almost all patients developed resistance and progressed to a recurrent CRPC stage[67,68]. This study addresses a vital question: why do certain

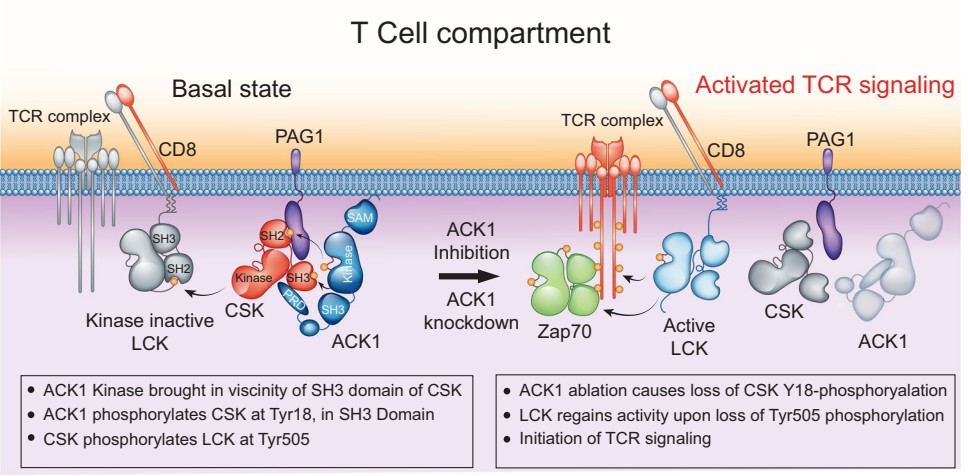

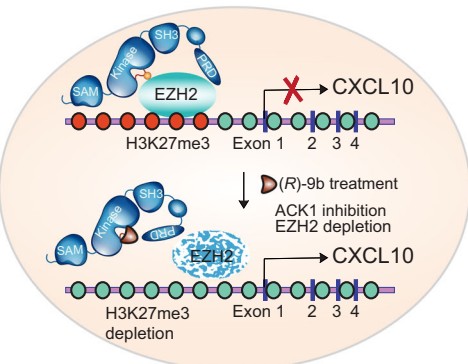

**Fig. 8 | A model for role of ACK1 in T cell activation and corresponding paracrine signaling in tumors, a graphical representation.** PAG Tyr317-phosphorylation brings SH2 domain of CSK kinase in proximity of PAG and ACK1, which phosphorylates CSK at Tyr18 in SH3 domain. The proline-rich region of ACK1 (PRD) interacting with SH3 domain of CSK could further stabilize the ACK1-CSK interaction. pY18-CSK phosphorylates LCK at Tyr505, preventing ligand-independent TCR activation. Genetic ablation or its inhibition of ACK1 by (R)-**9b** not only prevented CSK Y18-phosphorylation and LCK Y505-phosphorylation, but also allowed LCK to regain its kinase activity by its Y394-phosphorylation, subsequently causing activation of TCR signaling. In the tumor compartment, ACK1 inhibits the expression of the chemokine CXCL10 by depositing H3K27me3 marks recruiting EZH2 along the upstream region of CXCL10. Upon pharmacological inhibition of ACK1 by (R)-**9b**, EZH2 is depleted removing the inhibitory marks that enhances CXCL10 expression which could probably improve the active T-cell recruitment in the prostate tumor milieu.

small-molecule inhibitors perform poorly in clinical trials after they showed promising results in targeting cancer cells in in vivo studies? A plausible explanation emerged when a chimeric mouse model of mCRPC exhibited compromised T-cell activation due to inhibition of Src family kinases[69], revealing that CRPC-effective inhibitors are likely to be those that not only silence AR/AR-V7[68], but also do not compromise ability of T lymphocytes to attack cancer cells. (R)-**9b** is a compound (thus far the only one, to our knowledge) that not only inhibits AR/AR-V7 expression in mCRPCs[13] but also activates the host immune system to mount an effective response against cancer cells.

This report elucidates how the tumour microenvironment utilizes tyrosine kinase signalling as a molecular immune-rheostat that controls the immune response by simultaneously changing T-cell activation and T-cell-recruiting gene expression. These data reinforce the notion that the pharmacological inhibition of ACK1 kinase can fine-tune the ability of the immune system to recognize and eradicate tumour cells. Further studies will be performed to determine whether ACK1 inhibition leads to desirable outcomes in other immune-refractory solid tumours.

## Methods

### Cell lines

TRAMP-C2, HEK293T and Jurkat cells were obtained from ATCC. LAPC4 and C4-2B cells source and growth conditions are described earlier[13]. TRAMP-C2 were grown in DMEM containing 4 mM L-glutamine and 1.5 g/L sodium bicarbonate supplemented with 10 nM dehydroisoandrosterone 90%, 5 μg/ml bovine insulin, 5% FBS and 5% Nu-Serum IV. HEK293T cells and Jurkat cells were cultured in DMEM and RPMI-1640, respectively, supplemented with 10% FBS. All cultures were maintained with 50 units/ml of penicillin/streptomycin (Invitrogen) and cultured in 5% CO2 incubator. All cultures were tested for mycoplasma contamination every 2 months using the PCR Mycoplasma Test Kit I/C (PromoKine). Identities of all cell lines were confirmed by Short Tandem Repeat (STR) Profiling.

### Generation of *Tnk2* conditional KO mice using Cre-loxP recombination strategy

All animal experiments were performed using the standards for humane care in accordance with the National Institutes of Health (NIH) *Guide for the Care and Use of Laboratory Animals*. Mice studies were

performed according to IACUC protocols approved in writing by Washington University in St. Louis Department of Comparative Medicine (IACUC protocol nos. 20180247 and 20180259). All mice were co-housed with 3–5 mice per cage and maintained in a controlled pathogen-free/germ-free environment with a temperature of 20–23 °C, 12/12 h light/dark cycle, 50–60% humidity, and food and water provided ad libitum. The *Tnk2* gene (NCBI Reference Sequence: NM_016788; Ensembl: ENSMUSG00000022791) is located on Mouse chromosome 16. 16 exons are identified, with the ATG start codon in exon 2 and the TGA stop codon in exon 16 (Transcript: ENSMUST00000115124). The 5' loxP site is located in intron 2 and the 3' loxP site is inserted in intron 3. This region is followed by the frt-flanked PGK promoter driven SV40-*neo* gene, located in intron 2. Cre-mediated recombination between these two sites will result in deletion of part of intron 2 and all of exon 3 to generate a stop codon TGG due to splicing of exon 2 with exon 4, causing premature termination (Supplementary Fig. 1b). The targeting construct was electroporated into C57BL/6 (black) embryonic stem (ES) cells. Cells containing the correctly targeted allele were identified by PCR using a forward primer in the genome, outside the region of targeting and reverse primer in the *neo* gene. Neo and 5' loxP sites were introduced into the second intron along with a new BamHI site just after 3' loxP in intron 3. As a result digestion of genomic DNA from G418 resistant clones resulted in the appearance of a 9.8 kb and a 13.5 kb bands corresponding to the wildtype and *neo* inserted alleles respectively. 23% of the clones were tested positive by PCR and the clones were reconfirmed by southern blotting.

Two embryonic stem cells (ES) clones (BO4 and CO4) each containing a single targeted *Ack1* allele were microinjected into albino C57BL/6 (B6) blastocysts. Four mice with 40–75% of chimera were observed. Chimeric males were then mated with B6 albino females to screen for germ line transmission. Two black pups, likely to be floxed for *Ack1* or *Ack1*^flx/wt were obtained. *Ack1*^flx/wt mice were bred with EIIa-Cre mice to determine whether loss of *Ack1* leads to embryonic lethality. The adenovirus EIIa promoter directs expression of Cre recombinase in preimplantation mouse embryos and in nearly all tissues. No embryonic lethality was seen. We obtained *Ack1* heterozygous mice, which were subsequently interbred to obtain homozygous knockout mice (Supplementary Fig. 1c, d). The tail PCR was done to confirm genotypes of the mice. *Ack1* KO mice were backcrossed to C57BL/6 for at least 10 generations.

### ALT assay
ALT activity was determined in liver lysates from the WT and *Ack1* KO mice using the Alanine Transaminase Activity Assay Kit (Abcam), according to the manufacturer's recommendations. The experiment was done in triplicates and the mean value has been plotted.

### ANA assay
Blood serum was collected from the WT and *Ack1* KO mice and the ANA assay was done using Mouse anti-nuclear Antibody (IgG) ELISA Kit (CUSABIO, Wuhan, China) according to manufacturer's protocol.

### Histology and staining
Tissues collected from the WT and *Ack1* KO mice were fixed in paraformaldehyde and paraffin embedded. 5-μm sections were cut and stained with H&E and Picro-Sirius red. Based on the degree of severity, inflammation and fibrosis were evaluated by pathologist (C.W.).

### Docking
We used ClusPro[34], ACK1 (PDB-entry 4HZR)[33] as receptor and CSK (PDB-entry 1K9A)[32] as ligand. We generated restraints for docking from the co-crystal structure of insulin receptor kinase (IRK) with peptide (PDB-entry 1IR3)[35] by aligning IRK on ACK1 and calculating seven Cα-Cα distances between structurally conserved residues on ACK1 and the position of the substrate tyrosine bound to IRK. We performed multiple parallel docking runs while relaxing restraints from 2.5 Å to 5 Å. Docking solutions were manually inspected. We chose the solution with the most similar backbone geometry around Y18 to that of the substrate peptide observed in PDB-entry 1IR3. We aligned the structure of IRK (PDB-entry: 1IR3) on ACK1 kinase when complexed with CSK to position the ATP and two $Mg^{2+}$ ions. The complex was analyzed in COOT[36] and the side chain chi-angles of Tyr18 were changed to rotate the side chain hydroxyl group towards ATPγS. We analyzed the complex interface with the PISA server[70].

### Protein expression and crystallization
A baculovirus vector encoding hexahistidine-tagged ACK1 kinase domain[39] was used to transfect Sf9 insect cells using the Bac-to-Bac System (Invitrogen). After 3 rounds of amplification, the recombinant baculovirus was used to infect 600 ml of Sf9 cells at $2 \times 10^6$ cells/ml. After three days of infection, the cells were harvested and lysed by two passages through a French pressure cell. The lysis buffer (40 ml) contained 20 mM Tris (pH 8.0), 5 mM beta-mercaptoethanol, 10 μg/ml leupeptin, 10 μg/ml aprotinin, 1 mM PMSF, and 1 mM $Na_3VO_4$. The lysate was centrifuged at $40,000 \times g$ for 30 min, then filtered successively over 5.0 and 0.8 μm filters. The filtrate was applied to a 3 ml nickel-nitriloacetic acid column using a peristaltic pump. The column was washed first with 150 ml of buffer containing 20 mM Tris (pH 8.0), 2 mM imidazole, 0.5 M NaCl, 10% glycerol, 5 mM beta-mercaptoethanol, and 1 mM $Na_3VO_4$. The second wash was with 50 ml of 20 mM Tris (pH 8.0), 1 M NaCl. After a final wash with 20 mM Tris (pH 8.0), Ack1 was eluted with 20 mM Tris buffer containing 100 mM imidazole, 5 mM beta-mercaptoethanol, and 10% glycerol. Fractions containing Ack1 were stored at −80 °C. Kinase activity was confirmed by the phosphocellulose paper binding assay using [$^{32}$P]-ATP and a peptide based on the phosphorylation site of Wiskott-Aldrich Syndrome Protein (WASP).

The N-terminal His-tag was removed from ACK1 by cleavage with TEV protease. ACK1 was dialyzed at 4 °C against 25 mM HEPES (pH 7.5), 20 mM imidazole, 0.5 mM TCEP, 300 mM NaCl, 10% glycerol, in the presence of His-tagged TEV (20:0.5 ratio). In order to separate TEV protease and the cleaved His-tag peptide, the solution was applied to a second Ni-NTA column. Cleaved ACK1 was eluted using column buffer containing 100 mM imidazole, while the His-tag and TEV remained bound to the resin. The eluate was collected, concentrated and purified further on a Superdex 200 (16/60) Column. The Superdex 200 column was equilibrated using 25 mM Hepes (pH 7.7), 300 mM NaCl, 0.5 mM TCEP, 20 mM MgCl2, and 10% Glycerol. Peak fractions were collected and analyzed by SDS–PAGE. Purified ACK1 was concentrated to 4 mg/ml and stored in size exclusion chromatography buffer at −80 °C.

For the crystallization, 4 mg/mL of purified ACK1 in SEC buffer was used. Purified ACK1 kinase domain was crystallized using the hanging-drop vapor diffusion method at room temperature (18 °C). Protein was incubated with inhibitor (*R*)-**9b** stock solution in 1:2 protein – inhibitor molar ratio on ice for ½ h. Incubated complex mixture was spun at 12000 rpm for a short span of 10 min to separate precipitates and aqueous complex solution, before the crystallization drop set up. The best crystals for the complex were obtained in drops that were equilibrated with the reservoir solution consisting of 50 mM Bis-Tris (pH 6.5), 23% (w/v) polyethylene glycol 3350, 100 mM MgCl2, and 2.5% Glycerol in approximately 2 days. Crystals were flash frozen by direct immersion in liquid nitrogen using 20% (v/v) glycerol as the cryoprotectant in reservoir solution.

### Structure determination
X-ray diffraction data were collected at NSLS-2 AMX and processed using XDS[71] accounting for anisotropic diffraction. The structure was solved by molecular replacement using the structure of ACK1 kinase

domain (PDN-entry 4HZR)[33] as a search model in Phenix[72] using Phaser[73]. The model was built in COOTt[36] and refined in Phenix.

## Mice tumor studies

All animal experiments were performed using the standards for humane care in accordance with the NIH Guide for the Care and Use of Laboratory Animals. The WT and *Ack1* KO mice used for the experiments were in-bred as described above. The C57BL/6 mice (Strain code: 027, Stock Number: C57BL/6-027) used for the ACK1 inhibitor studies was purchased from Charles River Laboratories, USA. All mice were co-housed with 3–5 mice per cage and maintained in a controlled pathogen-free/germ-free environment with a temperature of 20–23 °C, 12/12 h light/dark cycle, 50–60% humidity, and food and water provided ad libitum. For all mice experiments, tumor volumes of 1500–1700 mm³ or end of treatment period, was considered as the humane end point. The maximum allowed weight loss was 10% of the body weight.

$1.2 \times 10^6$ TRAMP-C2 cells were suspended in 200 µl of PBS with 50% matrigel (BD Biosciences) and were implanted subcutaneously into the dorsal flank of 6- to 8-weeks old WT and *Ack1* KO mice. Tumor volumes were measured twice a week using calipers. Formation of tumors was monitored over an entire 9–11 week period. At the end of the study, when the tumor volumes were approximately 1500–1700 mm³, all mice were humanely euthanized by carbon dioxide inhalation, followed by cervical dislocation. Tumors were extracted and weighed. Additionally, spleen, thymus and femurs were collected and lymph nodes were drained for immunoblotting. Splenocytes and thymocytes were isolated for flow cytometry and tumors were used for further quantitative studies.

To study the antitumor efficacy of (*R*)-**9b**, TRAMP-C2 cells were implanted in 6- to 8-weeks old male C57BL/6 mice. 4 weeks postinjection of cells when tumors were palpable, mice were injected subcutaneously with (*R*)-**9b** (or vehicle) at the concentrations 24 mg/kg of body weight, five times a week, for 4 weeks. Tumor volumes were measured twice weekly using calipers. At the end of the treatment period, all mice were humanely euthanized by carbon dioxide inhalation, followed by cervical dislocation. The tumors were extracted and weighed. Splenocytes were isolated for flow cytometry and tumors were used for further quantitative studies.

## Antibody depletion studies

To determine T-cell mediated growth of TRAMP-C2 tumors, T-cell depletion experiment was performed. WT and *Ack1* KO male mice (6–8 weeks) were injected intraperitoneally with 250 µg/mouse αCD4 (GK1.5, BioXcell) or 250 µg/mouse αCD8β (53-5.8, BioXcell) or IgG (control). Three days post the antibody injection, mice were subcutaneously implanted with $1.5 \times 10^6$ TRAMP-C2 cells that were suspended in 200 µl of PBS with 50% matrigel (BD Biosciences). Following tumor implantation, mice were injected with the above antibodies intraperitoneally, once a week, for 5 weeks. Tumor growth was monitored and measured with calipers. At the end of the treatment period, all mice were humanely euthanized by carbon dioxide inhalation, followed by cervical dislocation. Tumors were extracted and weighed.

## Adoptive transfer experiment

To determine tumor specific T cell responses, TRAMP-C2 tumors were implanted subcutaneously in 10–12 weeks old NSG mice purchased from the Jackson Laboratory (Strain number: 005557, JAX stock #005557; IACUC protocol no 20-0383). T cells were purified from the splenocytes of WT and *Ack1* KO mice. Purified T cells ($1.5 \times 10^6$) resuspended in PBS were injected once a week from the 4th day after tumor implantation. After 6 weeks, mice were humanely euthanized by carbon dioxide inhalation followed by cervical dislocation. Splenocytes were isolated and lymph nodes were drained. The T cell phenotype and levels of SPAS-1 expression were assessed.

Splenocytes or cells from drained lymph nodes were stained with Live/Dead Aqua (1:800), anti-CD3 PECy7 (1:400), anti-CD8 APC (1:400), anti-CD4 Pacific blue (1:400), anti-CD44 PE (1:400) and anti-CD62L PerCP-Cy5.5 (1:400) or SPAS-1 PE (1:550) or control tetramer followed by flow cytometry. Tumors were excised. For the analysis of TILs, tumors were dissociated in digestion media and a single cell suspension was made. TILs were stained with the above-mentioned antibodies along with anti-CD45 APC Cy7 (1:300) followed by flow cytometry. The persistence of the transferred T cells was assessed by injecting CFSE-labeled T cells in the NSG mice and T-cell proliferation was assessed periodically by isolating lymphocytes from blood collected by sub-mandibular puncture, followed by flow cytometry up to 14 days post adoptive transfer.

## ICB combination studies

To evaluate the effect of (*R*)-**9b** as a combination drug with ICB, $1.5 \times 10^6$ TRAMP-C2 cells were suspended in 200 µl of PBS with 50% matrigel (BD Biosciences) and were implanted subcutaneously into the dorsal flank of 6- to 8-weeks old male C57BL/6 mice. Mice were injected intraperitoneally with combination of anti–CTLA-4 [clone 9H10; BioXCell] and anti–PD-1 [RMP1–14; BioXCell] (100 µg each) antibodies (ICB) on days 32, 34, and 41. In the second set, once the tumors reached approximately 125 mm³ in size, mice were injected with (*R*)-**9b** in 6% captisol in PBS at the concentration of 24 mg/kg of body weight, five times a week, for 4 weeks. The third set of mice received ICB antibodies in combination with subcutaneous injections of (*R*)-**9b** five times a week, for 4 weeks. Tumor volumes were measured twice a week using calipers. At the end of the treatment period, all mice were humanely euthanized by carbon dioxide inhalation, followed by cervical dislocation. Tumors were extracted and weighed. Spleen lysates were used for immunoblotting. Splenocytes were isolated and stained with Live/Dead Aqua (1:800), anti-CD4 Pacific blue (1:400), anti-CD25 FITC (1:250). Cells were then permeabilized and intracellular staining was done with anti-FoxP3 PE (1:250) followed by flow cytometry.

## Collection of human prostate tissue samples for organoid studies

Human prostate tumor and adjacent normal tissue samples were obtained following radical prostatectomy with patient's consent under the IRB-approved GU Banking Protocol (HRPO #201411135). Normal (i.e., far from the tumor site) and tumor tissue (i.e., the center core of cancerous lesion) were identified by magnetic resonance imaging (MRI) studies and pathology reports. Tissues were dissected as per the above identification by a pathologist within 30 min of surgery to avoid ischemia. Prostate tissue specimens were collected immediately on ice, and a 1 mm core was fixed in formalin, embedded to be reviewed again by the board-certified pathologist (CW), who assigns a Gleason score to all specimen collected using this procedure. The period of tissue storage was no longer than 1 hr from surgery to culture to maintain organoid viability. Prostate tissues were washed twice with cold 1X Phosphate buffered saline and dissected into equal 3–5 mm pieces for molecular profiling and organoid culture as described below.

## Generation of human prostate derived tumor organoids (PDTO) from fresh tissues

To generate normal and tumor organoids, fresh tissues of normal/tumor origin from radical prostatectomies were washed twice with 1X PBS and minced into approximately 0.1–0.5 mm diameter pieces with disposable sterile surgical scalpel blades. The minced tissues were transferred to 2 ml of collagenase containing cell dissociation media and incubated at 37 °C for 45 min with continuous gentle rotation. The Cell dissociation media was prepared as follows: A 5 mg of Collagenase (Collagenase Type II, Gibco, no. 17101-015) was added to 1 ml of basal advDMEM/F12 (Advanced DMEM/F12,

Invitrogen, no. 12634-034) containing penicillin/streptomycin (Gibco, no. 10378016), 10 mM HEPES (Invitrogen, no. 15630-056) and 2 mM GlutaMAX (Invitrogen, no. 35050-079) and 10 µM of ROCK inhibitor Y27632 (Sigma, no. Y0503) and filter sterilized and stored for 3 days at 4 °C[74].

The dissociated prostate tissues were centrifuged at 500 g for 5 min at 4 °C, and the pellet was washed with ice-cold basal advDMEM/F12 media. The pelleted dissociated tissues were trypsinized in 5 ml of TrypLE (Gibco, no. 12605-028) with Y-27632 for 20 min with intermittent pipetting every 10 min. Cell suspension was washed with 10 ml of ice-cold basal advDMEM/F12 media. The dissociated cell suspension was filtered with a 40-µm mesh filter. Cells were counted and $1 \times 10^4$ cells were resuspended in 75% Matrigel (Corning) and plated in a 40 µl drop in the middle of one well of a 24 well plate. The plate was turned upside down and incubated for 30 min at 37 °C and 5% $CO_2$. A 0.5 ml of complete human prostate organoids culture media containing B27-serum free (Life technologies), Nicotinamide and N-acetylcysteine (Sigma Aldrich), Noggin (PeproTech), R-Spondin (R&D systems-Cultrex®), p38 MAP kinase inhibitor SB202190 (Sigma-Aldrich), epidermal growth factor (EGF), fibroblast growth factor10, FGF-10 and FGF2 (PeproTech), TGFβ kinase/activin receptor-like kinase (ALK5) inhibitor A83-01(Sigma-Aldrich), prostaglandin E2 (Tocris Bioscience) and Y-27632 (Abmol Biosciences) culture media was added after the matrigel solidification. Cells were maintained under this condition till the PDTOs achieved 300-micron size. The media was replenished every 3–4 days to maintain the growth integrity of PDTOs and passaged regularly after 3–4 weeks.

Organoids were characterized; briefly, the tissues (the starting material) and the organoids were reviewed by a board-certified pathologist (histological examination post-H&E staining). In addition to histology, we performed quantitative RT-PCR analysis for well-known markers of prostate cancer, e.g. androgen receptor (AR), HOXB13, and PSA/KLK3. The tumor organoids showed a significant upregulation of these markers compared to normal, validating the pathologist's analysis.

## C4-2B spheroids
C4-2B cells were seeded at approximately 5000 cells per well (24-well plate) into 40 µl growth factor reduced and phenol red-free Matrigel (Corning). C4-2B spheroids were maintained in organoid culture media and used for further experiments.

## Co-culture of (R)−9b treated PBMCs with human 3D prostate model systems
To assess the cell killing effect of (R)-9b treated peripheral blood mononuclear cells (PBMCs) on human 3D prostate organoid model systems, PBMCs were treated with (R)-9b (1 µM) and the compound was washed. Post 24 h co-culturing with (R)-9b treated PBMCs, the PDTOs or C4-2B spheroids were treated with dispase II (STEMCELL Technologies), followed by incubation for 60 min at 37 °C to digest the Matrigel. PDTOs and C4-2B spheroids were washed twice with ice-cold basal advDMEM/F12 media. To the pellet, 10 µl of binding buffer with propidium iodide and Hoechst was added and incubated for 10 min at room temperature in the dark. The entire content was transferred to a microscopic slide, and fluorescent images were taken in EVOS M5000 imaging system (Thermofisher Scientific).

## Recombinant DNA transfection
HEK293T cells were transfected with control pcDNA3.1 (Vector) ACK1, cACK1, ACK-KD, LCK, CSK or CSK-Y18F using X-tremeGENE HP DNA transfection reagent (Sigma). Transfected cells were grown in DMEM with 10% fetal bovine serum for 48 hr and/or treated with inhibitors/drugs and harvested for immunoblotting as per experimental conditions.

## Generation and affinity purification of the pY18-CSK antibody
Two CSK peptides coupled to immunogenic carrier proteins were synthesized as shown below, and pTyr18-CSK antibodies were custom synthesized by GenScript, NJ.

CSK Phosphorylated peptide: IAK**pY**NFHGTAEQDL
CSK peptide: IAKYNFHGTAEQDL

Two rabbits were immunized twice with the phosphopeptide, and the sera from these rabbits was affinity purified. Two antigen-affinity columns were used to purify the phospho-specific antibodies. The first column was the nonphosphopeptide affinity column. Antibodies recognizing the unphosphorylated residues of the peptide bound to the column. The flow-through fraction was collected and then applied to the second column, the phosphopeptide column. Antibodies recognizing the phospho-Tyr residue bound to the column and were eluted as phospho-specific antibodies.

## Knockdown of ACK1 by siRNA Interference
Jurkat cells were transfected with Ack1-specific siRNA (Dharmacon RNA Technologies, Lafayette, CO) or control siRNA by using X-tremeGENE transfection reagent (Sigma-Aldrich). Cells were allowed to culture for 48 hr and then harvested for immunoblotting experiments.

## Immunoblot analysis
Spleen and thymus were collected from the WT and *Ack1* KO mice. Organs were homogenized and lysed by sonication in receptor lysis buffer (RLB) containing 20 mM HEPES (pH 7.5), 500 mM NaCl, 1% Triton X-100, 1 mM DTT, 10% glycerol, phosphatase inhibitors (50 mM NaF, 1 mM Na2VO4), and protease inhibitor mix (Roche). PBMCs from prostate cancer patients, TRAMP-C2, C4-2B, HEK293T and Jurkat cells were transfected and/or treated as per experimental requirement and were harvested and lysed by sonication in RLB. Lysates were quantitated and 20 to 100 µg of protein lysates were boiled in SDS sample buffer, size fractionated by SDS-PAGE, and transferred onto a PVDF membrane (GE Healthcare). After blocking in 3% bovine serum albumin (BSA), membranes were incubated with the following primary antibodies: anti-ACK1 (1:1000), anti-H3 (1:1000), Anti-PD-1(1:1000), anti-CSK (1:1000), anti-FLAG (1:1000), anti- phosphoTyrosine (pTyr) (1:500), anti-LCK (1:1000), anti-Ub (1:1000), anti-LCK pY505 (1:1000), anti-LCK pY394 (1:1000), anti-LAT pY132 (1:1000), anti-ZAP-70 pY319 (1:1000), anti-PLC-g pY783 (1:1000), anti-ACK1 pY284 (1:1000), anti-PAG1 (1:1000), anti-HA (1:1000), anti-EZH2 (1:1000), anti-Actin (1:9,000). Following three washes in PBS-T, the blots were incubated with horseradish peroxidase-conjugated secondary antibody. The blots were washed thrice and the signals visualized by enhanced chemiluminescence (ECL) system according to manufacturer's instructions (Thermo Scientific) using Invitrogen™ iBright™ FL1000 imaging system.

For immunoprecipitation studies, cell or organ tissue were lysed by sonication in RLB, the lysates were quantitated and 0.5 to 1 mg of protein lysate was immunoprecipitated using 2 µg of anti-CSK, anti-pTyr, anti-FLAG, anti-HA, anti-Myc or anti-EZH2 antibody coupled with protein A/G sepharose (Santacruz) overnight, followed by washes with RLB and PBS buffers. The beads were boiled in sample buffer and immunoblotting was performed as described above. Densitometric analysis using ImageJ software (ImageJ, NIH, USA) was performed for each representative immunoblot image and actin normalized relative fold change intensity for each lane is incorporated, wherever required.

## Mass spectrometry
HEK293T ($2.2 \times 10^6$) cells were co-transfected with ACK1 or vector. Post 48 hr of transfection, cells were processed for LC-MS/MS analysis. Samples were digested overnight with modified sequencing grade trypsin (Promega, Madison, WI), Glu-C (Worthington, Lakewood, NJ), or Arg-C (Roche,Switzerland). Phosphopeptides were enriched using

Phospho Select IMAC resins (Sigma). A nanoflow ultra high performance liquid chromatograph (RSLC, Dionex, Sunnyvale, CA) coupled to an electrospray bench top orbitrap mass spectrometer (Q-Exactive plus, Thermo, San Jose, CA) was used for tandem mass spectrometry peptide sequencing experiments. The sample was first loaded onto a pre-column (2 cm × 100 μm ID packed with C18 reversedphase resin, 5 μm, 100 Å) and washed for 8 min with aqueous 2% acetonitrile and 0.04% trifluoroacetic acid. The trapped peptides were eluted onto the analytical column, (C18, 75 μm ID × 50 cm, 2 μm, 100 Å, Dionex, Sunnyvale, CA). The 90-minute gradient was programmed as: 95% solvent A (2% acetonitrile + 0.1% formic acid) for 8 min, solvent B (90% acetonitrile + 0.1% formic acid) from 5% to 38.5% in 60 min, then solvent B from 50 to 90% B in 7 min and held at 90% for 5 min, followed by solvent B from 90 to 5% in 1 min and re-equilibrate for 10 min. The flow rate on analytical column was 300 nl/min. Sixteen tandem mass spectra were collected in a data-dependent manner following each survey scan. Both MS and MS/MS scans were performed in Orbitrap to obtain accurate mass measurement using 60 second exclusion for previously sampled peptide peaks. Sequences were assigned using Sequest (Thermo) and Mascot (www.matrixscience.com) database searches against SwissProt protein entries of the appropriate species. Oxidized methionine, carbamidomethyl cysteine, and phosphorylated serine, threonine and tyrosine were selected as variable modifications, and as many as 3 missed cleavages were allowed. The precursor mass tolerance was 20 ppm and MS/MS mass tolerance was 0.05 Da. Assignments were manually verified by inspection of the tandem mass spectra and coalesced into Scaffold reports (www.proteomesoftware.com).

## Immunophenotyping

Under sterile conditions, spleen, lymph nodes and femurs were harvested from naïve or TRAMP-C2 tumor bearing WT and *Ack1* KO mice. Single cells were made and RBCs were lysed using ACK lysis buffer. For immunophenotyping, $1 \times 10^6$ cells were incubated with Live/Dead Aqua (1:800), anti-CD3 PECy7 (1:400), anti-CD4 Pacific blue (1:400), anti-CD8 APC (1:400) antibodies to identify T-cells. The anti-CD19 PerCP-Cy 5.5 antibody (1:400) was used to identify B-cells, anti-NK1.1 PE antibody (1:500) was used to identify NK cells, anti-CD4 Pacific blue (1:400), anti-CD25 FITC (1:250) and intracellular anti-FoxP3 PE (1:250) antibodies were used to identify regulatory T cells, anti-CD11b PerCP-Cy 5.5 (1:400) and anti-Gr-1 APC (1:400) antibodies were used to identify subset of myeloid cells. Antibodies were incubated for 20 min, according to manufacturer's instructions (BD biosciences and BioLegend). T-cells were purified from splenocytes using mouse CD3⁺ T Cell Enrichment Column (R&D Systems) according to manufacturer's protocol. Purified T-cells or splenocytes were stained with Live/Dead Aqua (1:800) and either anti-CD3 PECy7 (1:400), anti-CD8 APC (1:400) and activation markers - anti-CD137 FITC (1:400), anti-CD44 PE (1:400) and anti-CD62L PerCP-Cy5.5 (1:400) or exhaustion markers - anti-PD1 FITC (1:400), anti-Lag3 PerCP-Cy5.5 (1:400), anti-Tim3 PE (1:400) or anti-CD4 Pacific blue (1:400) with activation marker anti-CD69 PE (1:400) antibodies. Cells were then permeabilized and intracellular staining was done with anti-perforin FITC (1:300), anti-IL2 APC-Cy7 (1:300) or anti-IFN gamma BV786 (1:300) antibodies. Tumor infiltrating T cells were identified by staining with Live/Dead Aqua (1:800), anti-CD45 APC-Cy7 (1:300), anti-CD3 PECy7 (1:400), anti-CD8 APC (1:400), anti-CD4 Pacific blue (1:400), anti-CD69 PE (1:400), anti-CD25 FITC (1:250) antibodies. Cells were then permeabilized and intracellular staining was done with anti-perforin FITC (1:300) and anti-FoxP3 PE (1:250) antibodies. Parallel immunophenotyping of spleen and tumor draining lymph nodes was performed in control and (*R*)-**9b** treated tumor bearing C57BL/6 mice staining with Live/Dead Aqua (1:800), anti-CD3, anti-CD4, anti-CD8, activation markers, exhaustion markers and intracellular staining with antibodies for markers as mentioned above. Samples were analyzed using BD FACSCanto II or LSR

Fortessa (BD Biosciences) and post-acquisition analysis was done using FlowJo software (Tree Star Inc).

Inhibition of pY18-CSK upon ACK1 knockdown or pharmacological inhibition was assessed by flow cytometry. Briefly, Jurkat cells treated with vehicle or (*R*)-**9b** and splenocytes from WT and *Ack1* KO mice were incubated with pTyr18-CSK primary antibody. Cells were washed, incubated with anti-rabbit Alexa Fluor® 488 antibody, fixed and flow cytometry was performed.

## Human PBMC separation

Whole blood was collected from healthy volunteers and prostate cancer patients in VACUTTE® tubes coated with sodium heparin. PBMC was isolated using Lymphocyte Separation Medium from Corning according to manufacturer's protocol.

## Intracellular calcium measurements

Jurkat cells were incubated with 1 μM (*R*)-**9b** for 3 h and 6 h and washed with PBS. Similarly, splenocytes isolated from C57BL/6 mice were treated with 1 μM (*R*)-**9b** for 6 h. Cells were loaded with Fluo-8 (Abcam) and incubated for 30 min at 37 °C. Intracellular calcium flux was measured using flow cytometry. Ionomycin was added at 300th second. Splenocytes and thymocytes isolated from WT and *Ack1* KO mice were incubated with TRAMP-C2 cells for 6 h and washed with PBS. Splenocytes from C57BL/6 mice were isolated and treated with (*R*)-**9b** for 6 h, washed and incubated with TRAMP-C2 cells for 6 h. Intracellular calcium flux measurement was done as described above. PBMCs isolated from healthy donors, CRPC and mHSPC patients were incubated with 1 μM (*R*)-**9b** for 6 h, washed and calcium flux was measured as described above. For the analyzing the response of (*R*)-**9b** treated cells upon anti-CD3 addition, dye-loaded cells were recorded for initial 30 s and then monitored after addition of anti-CD3 antibodies, followed by ionomycin addition.

## Quantitative RT-PCR

All RT reactions were done at the same time so that the same reactions could be used for all gene studies. For the construction of standard curves, serial dilutions of pooled sample RNA were used (50, 10, 2, 0.4, 0.08, and 0.016 ng) per reverse transcriptase reaction. One "no RNA" control and one "no Reverse Transcriptase" control was included for the standard curve. Three reactions were performed for each sample: 10 ng and a NoRT (10 ng) control. Real-time quantitative PCR analyses were performed using the ABI PRISM 7900HT Sequence Detection System (Applied Biosystems). All standards, the no template control (H₂O), the No RNA control, the no Reverse Transcriptase control were tested in six wells per gene (2 wells/plate x 3 plates/gene). All samples were tested in triplicate wells each for the 10 ng concentrations. The no RT controls were tested in duplicate wells. PCR was carried out with SYBR Green PCR Master Mix (Applied Biosystems) using 2 μl of cDNA and the primers in a 20-μl final reaction mixture. After 2-min incubation at 50 °C, AmpliTaq Gold was activated by 10-min incubation at 95 °C, followed by 40 PCR cycles consisting of 15 s of denaturation at 95 °C and hybridization of primers for 1 min at 55 °C. Dissociation curves were generated for each plate to verify the integrity of the primers. Data were analyzed using SDS software version 2.2.2 and exported into an Excel spreadsheet. The actin or 18s rRNA data were used for normalizing the gene values; i.e., ng gene/ng actin or 18s rRNA per well. The primer sequences are shown in Supplementary Table 2.

## Immunohistochemistry

Formalin-fixed paraffin-embedded (FFPE) tissue blocks were made from TRAMP-C2 tumors treated with Veh+IgG, ICB, (*R*)-**9b** and ICB + (*R*)-**9b**. Immunostaining was done on sections (4 μm) of the tissue blocks, which was mounted on glass slides and dried in an oven at 60 °C for 1 h. The antigen retrieval was done for 20 mins with

Tris based antigen retrieval solution (pH 9). Slides were incubated with Rabbit anti-mouse CD3 (E4T1B) antibody (Cell Signaling Technologies, USA). The mounted slides were analyzed under the bright field microscope and the total number of CD3 cells per field was calculated.

## RNA sequencing

C4-2B cells ($5 \times 10^7$ cells) were either treated with vehicle or (R)-**9b**. Total RNA was extracted using RNeasy kit from QIAGEN (Cat #74136). Total RNA integrity was determined using Agilent Bioanalyzer or 4200 TapeStation. Library preparation was performed with 5 to 10ug of total RNA with a Bioanalyzer RIN score greater than 8.0. Ribosomal RNA was removed by poly-A selection using Oligo-dT beads (mRNA Direct kit, Life Technologies). mRNA was then fragmented in reverse transcriptase buffer and heating to 94 degrees for 8 min. mRNA was reverse transcribed to yield cDNA using SuperScript III RT enzyme (Life Technologies, per manufacturer's instructions) and random hexamers. A second strand reaction was performed to yield ds-cDNA. cDNA was blunt ended, had an A base added to the 3' ends, and then had Illumina sequencing adapters ligated to the ends. Ligated fragments were then amplified for 12–15 cycles using primers incorporating unique dual index tags. Fragments were sequenced on an Illumina NovaSeq-6000 using paired end reads extending 150 bases. Base calls and demultiplexing were performed with Illumina's bcl2fastq software with a maximum of one mismatch in the indexing read. RNA-seq reads were then aligned to the Ensembl release 101 primary assembly with STAR version 2.7.9a1. Gene counts were derived from the number of uniquely aligned unambiguous reads.

## Chromatin immunoprecipitation (ChIP)

C4-2B and TRAMP-C2 cells ($5 \times 10^7$ cells) were either treated with vehicle or (R)−**9b**. Cells were harvested, fixed in 1% formaldehyde. Cell pellets were resuspended in RLB buffer and sonicated. The soluble chromatin was incubated at 4 °C with antibodies and protein-G and -A magnetic beads. The soluble chromatin was processed in the same way without immunoprecipitation and termed input DNA. The complexes were washed with RLB buffer, followed by ChIP buffer 1 and 2 (Active Motif), eluted with elution buffer and subjected to proteinase-K treatment. Fixed ChIP DNA was 'reverse' cross linked and subjected to proteinase-K treatment and purified using PCR DNA purification columns (Qiagen). The purified ChIP DNA was validated by real-time PCR as described above.

## ELISA

Levels of IFN-γ in the blood serum of WT and *Ack1* KO mice and cell culture supernatant of splenocytes from C57BL/6 mice treated overnight with (R)-**9b** was measured using mouse Interferon-γ ELISA Kit (R&D Systems) according to manufacturer's protocol.

## Cell mediated cytotoxicity assay

Splenocytes isolated from WT and *Ack1* KO mice were incubated with TRAMP-C2 cells. After 24 h, cells were labeled with 7-AAD (Biolegend) and the percentage of 7-AAD$^+$ cells were evaluated by flow cytometry. In addition, splenocytes were isolated from C57BL/6 mice, treated overnight with 1 μM (R)-**9b**. Cells were washed once with PBS and incubated with TRAMP-C2 cells, pre-stained with CFSE (BioLegend). After 24 h, cells were labeled with 7-AAD (Biolegend) and cell lysis was evaluated using flow cytometry analyzing the percentage of CFSE$^+$ 7-AAD$^+$ cells.

## Statistical analysis

All data are presented as mean ± SEM and all statistical parameters and analysis are mentioned in the figure legends respectively. Data for all experiments were analyzed with GraphPad Prism 8.0 software. All statistical analyses were performed using Student *t*-test unless otherwise specified. *p* values less than 0.05 were considered as statistically significant.

## Reporting summary

Further information on research design is available in the Nature Portfolio Reporting Summary linked to this article.

## Data availability

RNA-sequencing datasets used in this study can be found at:
https://www.ncbi.nlm.nih.gov/geo/query/acc.cgi
GEO accession **GSE211835**.
The structure ACK1 kinase domain with the compound (R)-**9b** can be found at:
**PDB ID** 7KP6 (Deposition ID D_1000252889).
Proteomics data has been submitted in PRIDE:
**Project accession:** PXD037546; **Project DOI:** https://doi.org/10.6019/PXD037546
Username: reviewer_pxd037546@ebi.ac.uk
Password: xIZA6hxV
The database will become public once the manuscript is accepted.
The data that support this study are available from the corresponding authors upon reasonable request. Source data are provided with this paper.

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

## Acknowledgements

N.P.M. is supported by NIH/NCI grants (1R01CA135328 and 5R01CA227025), Department of Defense (W81XWH-21-1-0202) and Prostate Cancer Foundation (PCF) grant (17CHAL06). N.P.M. also acknowledges Hamacher Family Prostate Cancer Research fund. F.Y.F. is supported by NIH/NCI grant (5R01CA227025). K.M. is recipient of Phi Beta Psi Sorority grant and Department of Defense award (W81XWH-21-1-0203). W.T.M. is supported by Dept. of Veterans Affairs Merit Award BX002292. Work in M.A.S. laboratory is supported by NIH R35 GM119437. W.T.M. is supported by NIH grant R01 AI164424 and VA # I01 BX002292. This research used beamlines AMX and FMX of the National Synchrotron Light Source II, a U.S. Department of Energy (DOE) Office of Science User Facility operated for the DOE Office of Science by Brookhaven National Laboratory under Contract No. DE-SC0012704. The Center for BioMolecular Structure (CBMS) is supported by NIH (P30GM133893), and by the DOE (KP1605010). M.A.S. and W.T.M. acknowledge the Stony Brook School of Medicine and the Office of the Vice President for Research. We thank Tiandao Li for GEO submission of RNA-sequencing data. We also thank John Koomen and Bin Fang, Moffitt Cancer Center Proteomics core for mass spectrometry analysis and PRIDE submission of proteomics data.

## Author contributions

D.S., S.C., K.M., S.B. performed the experiments; C.W. read the slides for pathologists analysis; M.K.T and M.S. performed crystallography studies; W.T.M, oversaw and guided purification of ACK1 protein; M.R. and E.K. provided human prostate tissue and blood samples and edited the manuscript. A.R. cultured spheroids. M.A.S. performed SPAS tetramer study; R.K.P. provided SPAS tetramers and analysed the data. F.Y.F analysed the data. N.P.M conceived the study, designed all the experiments; D.S., K.M. and N.P.M. wrote the manuscript. All authors reviewed the manuscript.

## Competing interests

N.P.M. and K.M. are named as inventors of two patents related to this work. Both the patents have been licensed by TechnoGenesys, Inc., N.P.M. and K.M are co-founders of TechnoGenesys, Inc., own stocks, and serve as consultants for TechnoGenesys, Inc. The remaining authors declare no competing interests.

## Additional information

¹Department of Surgery, Washington University at St Louis, St Louis, MO 63110, USA. ²Division of Urologic Surgery, Washington University at St Louis, St Louis, MO 63110, USA. ³Siteman Cancer Center, Washington University at St Louis, St Louis, MO 63110, USA. ⁴Division of Oncology, Department of Medicine, Washington University at St Louis, St Louis, MO 63110, USA. ⁵Anatomic and Clinical Pathology, Washington University at St Louis, St Louis, MO 63110, USA. ⁶Department of Pharmacological Sciences, Stony Brook University, Stony Brook, NY 11794, USA. ⁷Department of Physiology and Biophysics, School of Medicine, Stony Brook University, Stony Brook, NY 11794, USA. ⁸Department of Veterans Affairs Medical Center, Northport, NY 11768, USA. ⁹Department of Radiation Oncology, University of California, San Francisco, CA, USA. ¹⁰Helen Diller Family Comprehensive Cancer Center, University of California, San Francisco, CA, USA. ¹¹Division of Hematology and Oncology, Department of Medicine, University of California, San Francisco, CA, USA. ¹²Department of Urology, University of California, San Francisco, CA, USA. ¹³These authors contributed equally: Dhivya Sridaran, Kiran Mahajan. ✉e-mail: nupam@wustl.edu

