## [Peer Review File · Nature Communications]

Inhibiting ACK1-mediated phosphorylation of C-terminal Src kinase counteracts prostate cancer immune checkpoint blockade resistanceREVIEWER COMMENTS

Reviewer #1 (Remarks to the Author):

Sridaran et al address the role of the serine kinase ACK1 in the controls of T cells, and cytokine levels in tumor microenvironment. It is an interesting paper but several points need to be addressed. The paper is full of an astonishing number of different techniques but few are explained or executed rigorously. I would recommend a more focussed paper.

Specific Points

1) Unfortunately the paper is full of grammatical mistakes, for example on the first page of results 'staining with activation markers' should read staining with antibodies against activation markers'. 'by depriving B cells and macrophages' means what? Depleting these cells?, or 'in the activation of associated signalling molecules'..how do they know that they are activated? More rigor and precision is needed in the text.

2) Pg 4: 'practically indistinguishable' means what? Is there a difference to be seen? More information 'whole body knockdown' is needed...a line on how this was done? They refer to Fig. 1e related to CD44^{hi}CD62L^{lo} but not mention is made of naive, memory, resident memory cells etc. Extended Table1 is problematic: there is considerable variation between WT1 and WT2 samples making it difficult to know whether there differences between WT and KO exist or not? From example, CD24⁺Sca1⁺..looks like higher in KO but hard to know since WT1 has a value of 10.9 and WT.2 19.9. Ki67 is the same as is Shp2 and others. Further, these values are given as percentages...why not the MFI on expression...surely this could be the most revealing? The authors need to provide more information on which myeloid markers they used...I only see Ly6g⁺Grl⁺..

3) Fig. 1h-j: the range of cytokines is limited...is there an IL4 Th2 effect for example? Also odd that IFN levels are increased since AKT has been reported previously to be related to IFN production...are they arguing that other AKT family members, in particular AKT2 is responsible (AKT3 is mostly found in neuronal cells)? This result is puzzling to suggest that AKT negatively regulates IFN production...some explanation is needed>

4) It is unclear whether there is a phenotype in the thymus that might explain these results? Is thymus differentiation normal?

5) Figure 1: calcium data is hard to understand. They state 'basal levels'? Are they activating with anti-CD3 and at what concentrations and if not, why not?

6) Figure 2: much more information is needed on the docking analysis. Fig. 2c: again it is not clear why anti-CD3 activation was not included in this analysis? The reduction in pY505 and the increase in pY394 is convincing and interesting. The pY319 ZAP_70 blot is odd...also need to include MW markers. Why is there an increase in pY132 LAT in spleen but not thymus and what is this not matching pY783 for PLCgamma since PLC binds to the pY132 LAT site...this does not make any sense. Increase in CSK protein is interesting. Fig. 2d is interesting but again it would be important to test in the context of anti-CD3 activation or peptide TCR activation of KO cells crossed with a TCR transgenic such as OT1.

7) Fig, 2f is interesting and convincing. Lastly, the authors claim to have generated pY18-CSK specific antibodies but need further controls and titrations to show that the antibodies are specific....anti-phospho specific antibodies are remarkably non-specific at higher concentrations.

8) Figure 3: the ubiquitination work is poorly done. They need to show that the mutant cannot be ubiquitinated and a time course for WT vs Y18 mutant for CSK degradation.

9) Figure 5: the results are impressive but it is concerning the long period of time needed for tumour growth. The authors need to include another tumour model such as B16. Since the KO is global one wonders whether other cells are contributing to the tumour rejection although Fig. 6 partly addresses this issue. The authors need to look at the phenotype of the TILs and possibility tumor draining lymph nodes. Again the Fig 5 calcium experiment is odd...is there anti-CD3?

10) Figure 6: the cell transfer experiments are interesting...surprising that CD4+ T-cells suffice to regress tumours the same as CD8+ T-cells...this is certainly unexpected and unconventional. Are you looking at a loss of Tregs and are these CD4+ cells, cytolytic effectors? They exist but few people have reported such powerful effects similar to CD8+. Are the cells positive to GZMB and perforin? The authors need to explain these results.

11) Figure 11: it is not clear how organoids are being defined.

Reviewer #2 (Remarks to the Author):

This study demonstrates a link between intracellular signaling and potential for T cell responses in prostate cancer. The manuscript is well written and the experiments well executed. I believe that several key experiments will need to be conducted in order to strengthen their findings and the conclusions of this work.

Fig. 1:

The authors demonstrate robust knock out of ACK1 resulting in increased T cell activation (CD44 hi CD62L Lo and CD137+) in vivo and increased baseline calcium response in vitro. In addition IFN γ and IL-2 was increased.

However, it would be interesting to investigate the T cell phenotype and in vivo response further. Do T cells express conventional exhaustion markers such as PD-1, Lag3 and TIM3? Does the increased baseline activation result in any autoimmunity that can be measured by examining tissue damage by histology, serum IgG levels, serum chemistry.

Fig. 2:

The authors show data that ACK1 may target CSK while sparing LCK suggesting an ACK1-CSK-LCK interaction. Data is shown that deletion of c-term proline-rich domain (cACK1) ablates binding and phosphorylation of CSK. Is it specific to CSK, how does cACK1 change phosphorylation of other substrates? Does KO of CSK have the same effect as ACK1 KO?

In Fig. 2d-controls like cells expressing LCK and CSK without ACK1 would be convincing to strengthen the claim that ACK1 proline rich domain binds to CSK SH3 domain and ACK1 phosphorylates CSKY18.

Results with the (R)-9b ACK1 kinase inhibitor should be compared with ACK1 knock out using CRISPR in Jurkat cell lines.

Does ACK1 knock out or pharmacological inhibition also result in changed transcriptional activity through NFAT or NF κ B?

Fig.3

The authors demonstrate that ACK1 kinase activity regulates CSK activation and promotes proteasomal turnover to prevent excessive CSK activity.

Is this effected in activated T cells and does this effect downstream activation such as cytokine or proliferation?

Fig. 4

Data is shown to examine the effects on TRAMP-C2 tumor growth in Ack1 ko mice. The authors show that tumor growth is inhibited over time and that T cell activation is increased. However, T cell phenotypes were only examined after tumors were cleared and not intratumorally. What are changes in immune cells at earlier time points and within the tumor microenvironment? What is the phenotype of the T cells in terms of exhaustion markers such as PD-1, Lag3 and TIM3?

Fig. 5

TRAMP-C2 tumor growth was examined with Ack1 inhibitor (R)-9b treated mice showing similar response as was observed in Ack1 KO mice. Again it would be important to further characterize T cell phenotype and changes in exhaustion marker expression and activation such as cytokine expression and proliferation.

Using the inhibitor one could do intratumoral injection to determine if the same response can be achieved with local treatment.

Fig. 6

The authors show tumor growth inhibition with ACK1 KO or (R)-9b ACK1 kinase inhibitor is T cell dependent by depletion of CD4 and CD8 T cells and adoptive transfer into NSG mice. In the adoptive transfer experiments, injections were performed once a week suggesting the effect are short lived. It would be interesting to determine the persistence of treated vs. untreated T cells in this model.

Tetramer staining shows antigen specific response is increased in ACK1 KO but what is the phenotype of these T cells?

The investigators also examine the effects on tumor cells directly to show that EZH2/H3K27/me3 epigenetic signaling regulates CXCL10 expression by prostate tumors. It is not clear which type of CRPC cells were used in these studies. The authors suggest that increased CXCL10 expression recruits CXCR3+ T cells but this was not formally demonstrated with increased numbers of T cells in the tumor. This could be easily determined with follow up experiments.

FIG. 7

Lastly, the investigators examine ICB combination with ACK1 inhibition in vivo. ICB treatment robustly activates ACK1 resulting in CSK activation and LCK inhibition suggests ICB results in T cell activation feedback loop that can be counterproductive in cancer. Data is shown that T cells from an ICB treated cancer patient had increased pACK1, pY18CSK and pY505-LCK which respond to (R)-9b treatment. It would be important to show that these changes in signalling also translate to changes in T cell function and proliferation. How does this compare with response in healthy donor T cells?

In addition, it was shown that (R)-9b treatment increased prostate cancer PBMC killings of C4-2B spheroids or primary human prostate cancer organoids and this effect was enabled with PBMC treatment rather than organoid treatment. Given that the authors demonstrated tumor specific effects it would be good to compare T cell specific responses with tumor specific response (ie. treating tumor cells) or whole organoid treatment with both PBMC and organoid co-culture.

Reviewer #3 (Remarks to the Author):

The authors found a novel signaling function of ACK1-CSK to inactivate T cells using ACK1-knockout (KO) mouse model (Fig. 1-2) and demonstrated that the tumor growth of TRAMP-C2, mouse prostate cancer, was suppressed in ACK1-KO mouse (Fig. 4), or by ACK1 kinase inhibitor, (R)-9b (Fig. 5). Interestingly, anti-tumor activity of (R)-9b was inhibited by Immune-checkpoint blockade (ICB) (Fig 7a). Although the study is vigorous and contains new findings including a novel T cell activation signaling, there are some misinterpretations of the data and questions need to be solved.

Major issues:

1. Anti-tumor activity of (R)-9b was inhibited by ICB. This fact suggested the possibility that the combination of (R)-9b and ICB may induce and activate the regulatory T cells. Throughout the manuscript, the analysis of regulatory T cells (CD4+CD25+Foxp3+ cells) is missing.
2. The authors postulate that (R)-9b can induce the CXCL10 expression and change the tumor microenvironment (TME) more attractive to CTLs. However, the tumor growth inhibitory effect in ACK1-KO mouse model (WT-ACK1 tumor cells and KO-ACK1 T cells) is exactly same as that in (R)-9b treated mouse (ACK-1 kinase is inhibited both in tumor cells and T cells). If the postulation is correct, the antitumor activity in (R)-9b should be better than ACK1-KO mouse model. Nonetheless, in the manuscript, there is no data to confirm that ablation of ACK1 could induce the TME more attractive to T cells. Thus, the title of "Parallel Host T Cell Reinvigoration and Augmentation of Leukocyte Attractant by Cancer Cell Upon ACK1 inhibition Overcomes Immune Resistance" is not suitable to this study.
3. In the experiments of Fig. 4, 5, 7, TIL (tumor infiltrating lymphocytes) analysis by flowcytometry or immunohistochemical study of tumor tissues are needed to evaluate the TME.

Minor issues:

The data of tumor weight Fig. 4b, 5b and 7c are not necessary, they are overlapped with tumor volume data Fig. 4a, 5a, and 7a.

Point-by-point response to Reviewer's Comment

Reviewer #1 (Remarks to the Author):

Specific Points

1) Unfortunately the paper is full of grammatical mistakes, for example on the first page of results 'staining with activation markers' should read staining with antibodies against activation markers'. 'by depriving B cells and macrophages" means what? Depleting these cells?, or 'in the activation of associated signalling molecules"..how do they know that they are activated? More rigor and precision is needed in the text.

Response: We thank the reviewer for carefully reading our manuscript. This revised version has been thoroughly checked for grammatical mistakes and the sentences have been edited to the precise context, avoiding errors.

2) Pg 4: 'practically indistinguishable" means what? Is there a difference to be seen? More information 'whole body knockdown' is needed...a line on how this was done?

They refer to Fig. 1e related to CD44^{hi}CD62L^{lo} but not mention is made of naïve, memory, resident memory cells etc.

Extended Table1 is problematic: there is considerable variation between WT1 and WT2 samples making it difficult to know whether there differences between WT and KO exist or not? From example, CD24⁺Sca1⁺..looks like higher in KO but hard to know since WT1 has a value of 10.9 and WT.2 19.9. Ki67 is the same as is Shp2 and others.

Further, these values are given as percentages...why not the MFI on expression...surely this could be the most revealing?

The authors need to provide more information on which myeloid markers they used...I only see Ly6g⁺Grl⁺..

Response: The *Ack1* KO mice had no difference in the general appearance when compared to the WT mice. The sentence had been edited in the revised manuscript for clarity of information.

The generation of whole body *Ack1* KO has been elaborately explained in the methodology, however, the relevant information has been added to the results section in the revised manuscript, for better clarity.

We show data on naïve, central and effector memory T cells in **Supplementary Fig. 3a**. In brief, we see that there is an increase in the effector memory (EM) cells compared to the central memory (CM) cells in the KO mice. There is no change in the naïve cells.

We agree, there were variations in three markers, Ki67, pShp2 and Sca1. To validate whether there was indeed any change, we performed flow cytometry in 3 WT and 3 KO mice, please see the **Supplementary Fig. 3b-d**. We observed a significant decrease in the CD45⁺ pShp2⁺ cells in the *Ack1* KO mice, as compared to the WT mice. While the CD45⁺ Sca1⁺ cells were not significantly altered in WT vs KO mice. Further, we observed a marginal, but not statistically significant increase in the CD3⁺ Ki67⁺ cells in the *Ack1* KO mice, as compared to the WT mice (**Supplementary Fig. 3d**).

The percentages were derived from MFI, so it is unlikely to provide different information. We can include MFI as a raw data.

The Supplementary Table 1 was generated to analyze the myeloid markers and the specific signaling molecules, by CyTOF panel. In addition to Ly6g⁺Grl⁺, following four myeloid markers are shown in the table.

CD16/32 - marker for myeloid progenitor cells,

CD41 - marker to detect myeloid biased adult HSC cells (myeloid/erythroid cell marker)

CD150 - marker for of activated B cells

CD117 - myeloid blast marker

None of these four myeloid markers exhibited significant changes in WT versus KO mice.

3) Fig. 1h-j: the range of cytokines is limited...is there an IL4 Th2 effect for example? Also odd that IFN levels are increased since AKT has been reported previously to be related to IFN production.....some explanation is needed

Response: We have performed a comprehensive cytokine profile of the WT, *Ack1* KO and vehicle & (R)-9b treated mice. The data has been included as **Supplementary table 2**.

We measured the serum levels of 18 cytokines and chemokines. An increase in the levels of anti-tumorigenic cytokines or chemokines like IFN β -1, IFN- γ , IL-7, IL-9, IL-15, MIP-1 α and MIP-1 β was observed in serum samples from TRAMP-C2 injected *Ack1* KO mice (**Supplementary Table 2**). The cytokine profile of (R)-9b treated TRAMP-C2 tumor bearing mice too showed an increase in IFN β -1, IFN- γ , IL-7, IL-9, and IL-12p70 (**Supplementary Table 2**).

In addition, a decrease in the pro-inflammatory and pro-tumorigenic cytokines such as GM-CSF, IL-6, IL-17, MIP-3 α , IP-10, CXCL5, IL-11, IL-16, and MCP-5 was observed in blood serum samples of TRAMP-C2 injected *Ack1* KO mice (**Supplementary Table 2**). Further, (R)-9b treated TRAMP-C2 tumor bearing mice showed decrease in GM-CSF, IL-6, IL-17, MIP-3 α , IP-10 and M-CSF levels (**Supplementary Table 2**). A detail explanation for these data is provided in this revised manuscript.

Recent studies have shown that CD8⁺ T cells are divided into two distinct types: IFN- γ and IL-2 producing cells. The IL-2⁺ CD8⁺ T cells produce helper cytokines, while IFN- γ ⁺ CD8⁺ T cells produce cytotoxic molecules (Vrbensky JR, et. al. *Increased cytotoxic potential of CD8⁺ T cells in immune thrombocytopenia*. Br J Haematol. 2020;188(5):e72-e76). During infection and cancers, antigen-specific immunity drives the production of IFN- γ by CD8⁺ T lymphocytes and IFN- γ expression in T cells is known to be driven by T cell receptor mediated downstream signaling (Abd Hamid M. et., al. *Defective Interferon Gamma Production by Tumor-Specific CD8⁺ T Cells Is Associated With 5'Methylcytosine-Guanine Hypermethylation of Interferon Gamma Promoter*. Front Immunol. 2020; 5;11:310). These published data made us to examine IFN- γ levels in KO mice and (R)-9b treated mice, as an additional readout of T-cell activation.

4) It is unclear whether there is a phenotype in the thymus that might explain these results? Is thymus differentiation normal?

Response: We did not observe a striking change in the phenotype of the thymus and the thymus differentiation was normal.

5) Figure 1: calcium data is hard to understand. They state "basal levels"? Are they activating with anti-CD3 and at what concentrations and if not, why not?

Response: To demonstrate that loss of *Ack1* indeed causes T cell activation, which is reflected in calcium mobilization, we performed the experiment without addition of anti-CD3 antibodies, prior to calcium flux measurements. The splenocytes and thymocytes from KO mice exhibited higher calcium flux as compared to splenocytes and thymocytes isolated from WT mice (**Fig 1f** and **1g**). To further elaborate, we performed a calcium flux assay with addition of anti-CD3 antibody (concentration of 0.5µg/ml). As expected, splenocytes from both, WT and KO mice exhibited significant increase in calcium flux upon anti-CD3 antibody treatment (**Fig 1h**).

Similarly, splenocytes, thymocytes and Jurkat cells treated with ACK1 inhibitor (*R*)-**9b** exhibited significant increase in calcium flux (**Fig. 2g, 2h, 5d** and **5e**). Together, these data suggest that loss of *Ack1* activity causes T cells activation, which was similar to the increase seen upon addition of anti-CD3 antibody.

The term “basal” was used to define this difference between the WT and *Ack1* KO in naïve situations. However, the sentence has been edited in the revised manuscript to ensure clarity of information.

6) Figure 2: much more information is needed on the docking analysis.

Fig. 2c: again it is not clear why anti-CD3 activation was not included in this analysis?

The reduction in pY505 and the increase in pY394 is convincing and interesting. The pY319 ZAP_70 blot is odd..., also need to include MW markers. Why is there an increase in pY132 LAT in spleen but not thymus and what is this not matching pY783 for PLCgamma since PLC binds to the pY132 LAT site....

Increase in CSK protein is interesting. Fig. 2d is interesting but again it would be important to test in the context of anti-CD3 activation

Response: We have elaborated the docking analysis. More information is included in Methods section.

The calcium response studies clearly indicated that loss of ACK1 or its kinase activity by small molecule inhibitor resulted in T-cell activation, and addition of anti-CD3 antibody is not needed. That is why Fig. 2c was performed wherein spleen & thymus from ACK1 KO mice were examined and significant activation of various downstream signaling entities was noticed.

However, to further understand role *Ack1* in T cell activation, Jurkat cells were treated with anti-CD3 antibody and immunoblotting was performed, shown in **Supplementary Fig. 4a**. Upon anti-CD3 antibody treatment, there was a rapid decrease in ACK1 activation (loss of pY284-ACK), reflected in loss of pY505-LCK levels, and concomitant increase in pY394-LCK levels. These data suggests that loss of ACK1 activity is very early and crucial step in T cell activation.

We rerun the lysates, followed by immunoblotting for pY319-ZAP70, pY132-LAT and pY783-PLCγ. Please see the new blots in Fig 2C. The MW markers are included.

Since, CSK levels are dependent upon its polyubiquitination post-phosphorylation, based on data shown in **Figure 2f** and **Fig. 3e**, we performed anti-CD3 antibody treatment of Jurkat cells. A significant increase in CSK levels was seen post anti-CD3 antibody treatment (**Supplementary Fig. 4b**).

7) Fig, 2f is interesting and convincing. Lastly, the authors claim to have generated pY18-CSK specific antibodies but need further controls and titrations to show that the antibodies are specific....anti-phospho specific antibodies are remarkably non-specific at higher concentrations.

Response: As per the reviewer's suggestion, a detailed characterization of our antibody with more controls have been done, and added to **Supplementary Fig. 6a-d** in revised manuscript. Briefly, dot blot analysis revealed that these antibodies recognized the CSK peptide with the Y18-phosphorylated residue at center (pY18-CSK phosphopeptide), but the antibodies did not recognize a CSK peptide in which Tyr18 is not phosphorylated or one containing an unrelated phosphopeptide (pY37-H2B) (**Supplementary Fig. 6a**). In addition, when these antibodies were incubated with pY18-CSK phosphopeptide and then used for immunoblotting, the antibodies were neutralized and no signal was observed (**Supplementary Fig. 6b**, middle panel). In contrast, when these antibodies were incubated with unrelated phosphopeptide (derived from histone H2B), a robust signal was observed (**Supplementary Fig. 6b**, top panel). In addition, antibodies against pY18-CSK were assessed against multiple phosphopeptides, derived from phospho-AKT, phospho-ATP synthase, phospho-histones H2A, H2B, H3 and H4. The antibodies did not cross react with any of these phospho-peptides (**Supplementary Fig. 6c**). Further, treatment of cells with phosphatase inhibitor cocktail significantly increased pY18-CSK levels (**Supplementary Fig. 6d**, top panel).

8) Figure 3: the ubiquitination work is poorly done. They need to show that the mutant cannot be ubiquitinated and a time course for WT vs Y18 mutant for CSK degradation.

Response: The ubiquitination experiment in the CSK Y18F mutated condition and its degradation has been done and included in the revised manuscript, please see **Fig. 3d** and **3e**. Briefly, to explore temporal regulation of CSK levels, HEK 293T cells were co-transfected with ACK1, CSK or CSK-Y18F mutant and were cultured in Cycloheximide for 0, 2, 4 and 6 hrs. Almost complete loss of CSK was seen in 6 hours, however, CSK Y18F-mutant was not degraded (**Fig. 3e**).

9) Figure 5: the results are impressive but it is concerning the long period of time needed for tumour growth. The authors need to include another tumour model such as B16.

Since the KO is global one wonders whether other cells are contributing to the tumour rejection although Fig. 6 partly addresses this issue. The authors need to look at the phenotype of the TILs and possibility tumor draining lymph nodes.

Again the Fig 5 calcium experiment is odd...is there anti-CD3?

Response: Prostate tumors are known to be slow growing tumors, and clinically it takes a relatively long time for formation of detectable lesions. Further, an important reasons for treatment failure in prostate cancers is the late development of castration resistant tumors (CRPC), which is mostly lethal. Our study was aimed to recreate a syngeneic xenograft model that could mimic the clinical condition of Prostate cancer patients and hence we chose TRAMP-C2 cells for the induction of tumors in our experimental mice. This is a well-established prostate tumor model, used extensively by Dr. Allison himself, a pioneer in ICB studies (Waitz R, Solomon SB, Petre EN, Trumble AE, Fassò M, Norton L, Allison JP. Potent induction of tumor immunity by combining tumor cryoablation with anti-CTLA-4 therapy. *Cancer Res.* 2012 Jan 15;72(2):430-9). An important aspect of this model is a well-established growth pattern of prostate tumors, which supports an elaborate assessment of immune modulation during prostate tumor progression over a period. In addition, TRAMP-C2 tumors often respond poorly to androgen-deprivation strategies and thus behave like CRPC tumors.

Another publication which provides further insight into importance of this model to study immune regulation in response to prostate cancer is as follows:

Garcia-Hernandez Mde L, Gray A, Hubby B, Klinger OJ, Kast WM. Prostate stem cell antigen vaccination induces a long-term protective immune response against prostate cancer in the absence of autoimmunity. *Cancer Res.* 2008 Feb 1;68(3):861-9.

We had also tried the B16 tumors for our study, as an additional model. Owing to a rapid proliferation of B16 tumors, which impedes sustained analysis over considerable duration, we continue to use prostate tumor model. Additional reason to use prostate tumor model was the relative paucity of such studies in prostate cancer field. B16 tumors data is shown in **Supplementary Fig. 8d** in revised manuscript. Consistent with data shown in Fig. 4a, a significant decrease in B16 tumors was seen in the *Ack1* KO mice

We have done the immunophenotyping of the tumor draining lymph nodes as suggested by the reviewer and included in the revised manuscript. Data is shown in **Supplementary Fig. 7f, 10b and 11b**.

As described above, inhibition of ACK1 kinase activity could result in T cell activation. Accordingly, calcium flux was done isolating the splenocytes and thymocytes and treating them with ACK1 inhibitor, (*R*)-**9b** in the absence and presence of anti-CD3 antibody (please see **Fig. 2g** and **h**). (*R*)-**9b** treatment significantly increased the calcium flux (**Fig. 2g**); however, further stimulation with anti-CD3 antibody did not augment the magnitude of maximal calcium peaks (**Fig. 2h**).

10) Figure 6: the cell transfer experiments are interesting...surprising that CD4⁺ T-cells suffice to regress tumours the same as CD8⁺ T-cells...this is certainly unexpected and unconventional.

Are you looking at a loss of Tregs and are these CD4⁺ cells, cytolytic effectors? They exist but few people have reported such powerful effects similar to CD8⁺. Are the cells positive to GZMB and perforin? The authors need to explain these results.

Response: Our results show that CD4 cells also play a role in controlling prostate tumor growth. Published data indicates that prostate tumor microenvironment in both, mouse (prostate tumors derived from the TRAMP mice) and human, express MHC class I and II (Nanda NK, Birch L, Greenberg NM, Prins GS. MHC class I and class II molecules are expressed in both human and mouse prostate tumor microenvironment. *Prostate.* 2006, 66(12):1275-84). More cases are being reported in literature; in a recent issue of *Nature*, analysis of CAR T cells that persist for ten years in CLL patients was performed. They find a dominant cytotoxic CD4⁺ population, raising the possibility that CD4⁺ T cells play an important role in durable T cell therapy (Melenhorst J.J. et al. Decade-long leukaemia remissions with persistence of CD4⁺ CAR T cells. *Nature.* 2022; 602: 503-509). These long-persisting CD4⁺ T cells exhibited cytotoxic characteristics along with ongoing functional activation and proliferation.

Yes, loss of Tregs was observed in KO mice (**Supplementary Fig. 8b**), and was also seen upon (*R*)-**9b** treatment (**Supplementary Fig.15**)

In contrast, an increase in CD4⁺ CD69⁺ Perforin⁺ cells was observed in KO mice (**Supplementary Fig. 8a**). In addition, pharmacological inhibition by (R)-9b also resulted in increased CD4⁺ CD69⁺ Perforin⁺ cells (**Supplementary Fig. 10a**).

11) Figure 11: it is not clear how organoids are being defined.

Response: Organoids are small, self-organized three-dimensional tissue cultures that are derived from stem cells or Cancer stem-like cells (CSC) present in patient biopsies. Organoids often replicate the complexity of an organ. We (and others) have successfully grown prostate organoids from fresh prostate tissue specimens obtained from patients. How organoids were generated from prostate cancer patients and characterized is described in 'Methods' sections.

Briefly, the tissues (the starting material) and the organoids were reviewed by a board-certified pathologist (histological examination post H&E staining). In addition to histology, we performed quantitative RT-PCR analysis for well-known markers of prostate cancer, e.g. Androgen receptor (AR), HOXB13, and PSA/KLK3. The tumor organoids showed a significant upregulation of these markers compared to normal, validating the pathologist's analysis.

Reviewer #2 (Remarks to the Author):

This study demonstrates a link between intracellular signaling and potential for T cell responses in prostate cancer. The manuscript is well written and the experiments well executed. I believe that several key experiments will need be conducted in order to strengthen their findings and the conclusions of this work.

Response: Thank you so much. We have diligently performed experiments addressing all the questions.

Fig. 1: The authors demonstrate robust knock out of ACK1 resulting in increased T cell activation (CD44 hi CD62L Lo and CD137+) in vivo and increased baseline calcium response in vitro. In addition IFN γ and IL-2 was increased. However, it would be interesting to investigate the T cell phenotype and in vivo response further. Do T cells express conventional exhaustion markers such as PD-1, Lag3 and TIM3?

Does the increased baseline activation result in any autoimmunity that can be measured by examining tissue damage by histology, serum IgG levels, serum chemistry.

Response: We analyzed the expression of the exhaustion markers in the WT and *Ack1* KO mice injected with TRAMP-C2 cells. A decrease in PD-1, Lag3 and Tim3 was observed in tumor infiltrating lymphocytes or TILs (**Supplementary Fig. 8c**). Similarly, pharmacological inhibition of ACK1 by (R)-9b too caused a decrease in PD-1, Lag3 and Tim3 (**Supplementary Fig. 11a and 11b**).

The reviewer has rightly pointed out the potential possibility of autoimmunity in the KO mice due to the increased T cell activation threshold. We assessed if autoimmunity can be detected in organs like liver, kidney, small intestine and lungs using histology and picrosirius red staining. We also assessed the liver function with ALT assay and occurrence of anti-nuclear antibodies in the blood serum of WT and KO mice. Alanine Transaminase (ALT) assay of the WT and KO mice showed no difference, indicated absence of tissue damage due to autoimmunity (**Supplementary Fig. 4e**). Similarly, there was no change in the levels of anti-nuclear antibodies (ANA) between the WT and KO mice (**Supplementary Fig. 4f**). Moreover, histological analysis of the organs stained with H&E and Picrosirius red from older WT and KO mice too showed minimal signs of inflammation

and fibrosis (**Supplementary Fig. 4g** and **4h**). Taken together, these data indicates that although ACK1 kinase loss could activate T cells response, it does not lead to hyper-reactive T cells to generate robust autoimmune disease. However, we do not rule out the possibility that *Ack1* KO mice derived T cells could potentially be more prone to mount a destructive immune response against autoantigens after viral infection or other immune stresses.

We would also like to share some confidential information, we became recently aware of unpublished data wherein two mutants that inactivates ACK1 kinase activity were identified in patients with Systemic Lupus Erythematosus (SLE). This provides some insight into role of ACK1 in immune regulation, however, more detailed studies are needed to make clear distinction between ACK1's role in anti-cancer immunity and predisposition to SLE.

Fig. 2: The authors show data that ACK1 may target CSK while sparing LCK suggesting an ACK1-CSK-LCK interaction. Data is shown that deletion of c-term proline-rich domain (cACK1) ablates binding and phosphorylation of CSK. is it specific to CSK, how does cACK1 change phosphorylation of other substrates? In Fig. 2d-controls like cells expressing LCK and CSK without ACK1 would be convincing to strengthen the claim that ACK1 proline rich domain binds to CSK SH3 domain and ACK1 phosphorylates CSK Y18.

Does KO of CSK have the same effect as ACK1 KO?

Results with the (*R*)-**9b** ACK1 kinase inhibitor should be compared with ACK1 knock out using CRISPR in Jurkat cell lines.

Does ACK1 knock out or pharmacological inhibition also result in changed transcriptional activity through NFAT or NFκb?

Response: We thank the reviewer for the valuable suggestion that brings more clarity. As suggested, we performed immunoblotting of the cells expressing LCK and CSK, but without ACK. Please see revised **Fig. 2d** (right panel); in absence of ACK1, no changes in phosphorylation of other substrates were noticed.

Mice homozygous for disruption of the *csk* gene die *in utero* at 9 - 10 days of gestation, exhibiting defects in neurulation. Thus, a transgenic mice expressing a mutant form of CSK (CskAS) was developed. These mice express mutant CSK, which contains a point mutation at position 266 resulting in a threonine to glycine substitution. This mutation creates a larger ATP-binding pocket, allowing for specific inhibition by an analog of the kinase inhibitor PP1, 3-iodo-benzyl-PP1 (3-IB-PP1). CskAS mice have 2.5-fold as much expression of CSK as wild-type mice and display normal T cell development. However, inhibition of CSK with 3-IB-PP1 in thymocytes induces activation of Src Family of Kinases and proximal TCR signaling. Indeed, CskAS mice exhibited enhanced immune responses to very weak antigens, including antigens that might not even activate T cells under normal circumstances (Manz, B. N. et al. Small molecule inhibition of Csk alters affinity recognition by T cells. *Elife* 4, doi:10.7554/eLife.08088, 2015). Thus, the CSK loss does have the similar effect as ACK1 KO. We have included this information in Introduction section.

The results of pharmacological inhibition of ACK1 have been compared with ACK1 knockdown by siRNA in Jurkat cells and the data has been added to **Fig. 3f**. A significant loss of ACK1 Y284-, CSK Y18-, and LCK Y505-phosphorylation was observed, in addition, increased LCK Y394-phosphorylation was also seen (**Fig. 3f**).

We checked the NF- κ B transcriptional activation by qRT-PCR during our initial screening and observed that there was an increase in NF- κ B levels upon treatment with (R)-9b, which affirmed the possibilities of ACK1 influence in T-cell activation. The data has been included in **Figure 2i** in the revised manuscript.

Fig. 3: The authors demonstrate that ACK1 kinase activity regulates CSK activation and promotes proteasomal turnover to prevent excessive CSK activity. Is this effected in activated T cells and does this effect downstream activation such as cytokine or proliferation?

Response: Jurkat cells treated with anti-CD3 antibody exhibit clear signature of T cells activation; a rapid decrease in pY505-LCK levels in 5 minutes, and concomitant increase in pY394-LCK levels (**Supplementary Fig. 4a**). Further, longer CD3 treatment revealed rebuilding of CSK levels (**Supplementary Fig. 4b**). Consistent with these data, KO mice spleen and thymus lysates too exhibited decrease in pY18-CSK and pY505-LCK levels, and concomitant increase in pY394-LCK and total CSK levels (**Fig. 2c**). Flow cytometric analysis of pY18-CSK was also done and has been included in **Supplementary Fig. 6g** and **6h** in revised manuscript.

Consequently, a significant increase in IL-2 and IFN- γ is seen in T cells and has been included in **Supplementary Fig. 4c, 4d, 10c** and **10d**.

Fig. 4: Data is shown to examine the effects on TRAMP-C2 tumor growth in *Ack1* ko mice. The authors show that tumor growth is inhibited over time and that T cell activation is increased. However, T cell phenotypes were only examined after tumors were cleared and not intratumorally. What are changes in immune cells at earlier time points and within the tumor microenvironment? What is the phenotype of the T cells in terms of exhaustion markers such as PD-1, Lag3 and TIM3?

Response: The TILs were analyzed in the TRAMP-C2 tumors from WT and *Ack1* KO mice. We observed that there was a significant increase in the cytolytic T cells and a decrease in the Treg levels. The data has been included in **Supplementary Fig. 8a** and **8b**.

With respect to the exhaustion markers, there was a decrease in the expression of PD-1, Lag3 and Tim3 in the TILs. The data is provided in **Supplementary Fig. 8c**.

Fig. 5: TRAMP-C2 tumor growth was examined with *Ack1* inhibitor (R)-9b treated mice showing similar response as was observed in *Ack1* KO mice. Again it would be important to further characterize T cell phenotype and changes in exhaustion marker expression and activation such as cytokine expression and proliferation.

Using the inhibitor one could do intratumoral injection to determine if the same response can be achieved with local treatment.

Response: Yes. The phenotype of the T-cells, analysis of the exhaustion markers and cytokine profiling have been done and added to the revised manuscript in **Supplementary Fig. 10, 11** and **12** and **Supplementary Table 2**.

A significant increase in CD137⁺/CD8⁺ cytotoxic T cells, effector CD44^{hi}CD62L^{low} CD8⁺ T cells (**Fig. 5b, 5c**, and **Supplementary Fig. 9c** and **9d**) and CD4⁺ CD69⁺ cells expressing perforin (**Supplementary Fig. 10a**) in the (R)-9b-injected mice compared with vehicle-injected mice. The tumor draining lymph nodes also showed increased effector CD44^{hi}CD62L^{low} CD8⁺ T cells in the (R)-9b-injected mice (**Supplementary Fig. 10b**). Flow cytometric analysis also confirmed the increased levels of IL-2 and IFN- γ expression by the T-cells in (R)-9b treated mice

(**Supplementary Fig. 10c and 10d**). There was a decrease in the expression of exhaustion markers (PD-1, Lag3, Tim3) on the CD8⁺ T cells (**Supplementary Fig. 11a and 11b**).

Also, we performed the intra-tumoral injections of our inhibitor and observed similar tumor growth inhibition of TRAMP-C2 tumors. The data has been added, please **Supplementary Fig. 9a**.

Fig. 6: The authors show tumor growth inhibition with ACK1 KO or (R)-9b ACK1 kinase inhibitor is T cell dependent by depletion of CD4 and CD8 T cells and adoptive transfer into NSG mice. In the adoptive transfer experiments, injections were performed once a week suggesting the effect are short lived. It would be interesting to determine the persistence of treated vs. untreated T cells in this model.

Tetramer staining shows antigen specific response is increased in ACK1 KO but what is the phenotype of these T cells?

The investigators also examine the effects on tumor cells directly to show that EZH2/H3K27/me3 epigenetic signaling regulates CXCL10 expression by prostate tumors. It is not clear which type of CRPC cells were used in these studies. The authors suggest that increased CXCL10 expression recruits CXCR3⁺ T cells but this was not formally demonstrated with increased numbers of T cells in the tumor. This could be easily determined with follow up experiments.

Response: The persistence of the adoptively transferred T cells from *Ack1* KO and WT mice was assessed by injecting into the TRAMP-C2 tumor bearing NSG mice. Periodic sub-mandibular blood collection and flow cytometry of lymphocytes showed that the T cells from the KO mice had an improved homing and proliferation compared to the T cells from the WT mice (**Supplementary Fig. 14a and b**).

The phenotype of the adoptively transferred T-cells has been analyzed by flow cytometry and included in **Fig. 6c, 6d** and **Supplementary Fig. 13e**. The splenocytes revealed an increase in the percentage of effector CD8⁺ cells in the mice injected with T cells from *Ack1* KO mice, compared to the mice injected with WT T cells (**Fig. 6c** and **Supplementary Fig. 13e**). Analysis of TILs from the tumors of these NSG mice showed a significant increase in infiltrated lymphocyte in tumors from mice that were injected with T cells from *Ack1* KO mice (**Supplementary Fig. 13d**) and the phenotype of TILs showed an increased number of cells with CD44^{hi}CD62L^{low} expression (**Fig. 6d** and **Supplementary Fig. 13e**, lower panels).

The TRAMP-C2 cells poorly respond to AR antagonists and thus are considered to be the CRPC type. A significant increase in *Cxcr3* mRNA expression was observed in TILs obtained from mice treated with (R)-9b (**Supplementary Fig. 14c**).

FIG. 7: Lastly, the investigators examine ICB combination with ACK1 inhibition in vivo. ICB treatment robustly activates ACK1 resulting in CSK activation and LCK inhibition suggests ICB results in T cell activation feedback loop that can be counterproductive in cancer. Data is shown that T cells from an ICB treated cancer patient had increased pACK1, pY18CSK and pY505-LCK which respond to (R)-9b treatment. It would be important to show that these changes in signalling also translate to changes in T cell function and proliferation. How does this compare with response in healthy donor T cells?

In addition, it was shown that (R)-9b treatment increased prostate cancer PBMC killings of C4-2B spheroids or primary human prostate cancer organoids and this effect was enabled with PBMC

treatment rather than organoid treatment. Given that the authors demonstrated tumor specific effects it would be good to compare T cell specific responses with tumor specific response (ie. treating tumor cells) or whole organoid treatment with both PBMC and organoid co-culture.

Response: We assessed the response in healthy donor T-cells upon (R)-9b treatment. The immunoblotting showed that (R)-9b caused decrease in pY18-CSK and significant increase in pY3945-LCK and pY319-Zap70 (**Supplementary Fig. 17c**). This is reflected in improved calcium responses as shown in **Supplementary Fig. 17d**.

While studying the cytotoxicity of prostate cancer spheroids, we first designed our experiment where the C4-2B spheroids were co-cultured with PBMCs in the presence of (R)-9b and observed that there was an induction of cell death in the spheroids which led us to design the co-culture experiment in such a way we could figure out whether the cell killing was particularly due to the T-cell activation by (R)-9b. The data is included in **Supplementary Fig. 16d**.

Reviewer #3 (Remarks to the Author):

The authors found a novel signaling function of ACK1-CSK to inactivate T cells using ACK1-knockout (KO) mouse model (Fig. 1-2) and demonstrated that the tumor growth of TRAMP-C2, mouse prostate cancer, was suppressed in ACK1-KO mouse (Fig. 4), or by ACK1 kinase inhibitor, (R)-9b (Fig. 5). Interestingly, anti-tumor activity of (R)-9b was inhibited by Immune-checkpoint blockade (ICB) (Fig 7a). Although the study is vigorous and contains new findings including a novel T cell activation signaling, there are some misinterpretations of the data and questions need to be solved.

Response: We really appreciate the time taken by reviewer to carefully look through our manuscript and provide excellent feedback.

Major issues:

1. Anti-tumor activity of (R)-9b was inhibited by ICB. This fact suggested the possibility that the combination of (R)-9b and ICB may induce and activate the regulatory T cells. Throughout the manuscript, the analysis of regulatory T cells (CD4⁺CD25⁺Foxp3⁺ cells) is missing.

Response: This was very useful suggestion. We performed the experiment and were surprised to observe that administration of ICB with (R)-9b increased the population of regulatory T-cells when compared to the (R)-9b alone group. This could be a limiting factor for many other combination treatments with ICB. The data is now presented in **Fig. 7e** and **Supplementary Fig. 15**. Also, the analysis of Tregs has been done in all the experiments and has been added to **Supplementary Fig. 2c** and **8b**.

2. The authors postulate that (R)-9b can induce the CXCL10 expression and change the tumor microenvironment (TME) more attractive to CTLs. However, the tumor growth inhibitory effect in ACK1-KO mouse model (WT-ACK1 tumor cells and KO-ACK1 T cells) is exactly same as that in (R)-9b treated mouse (ACK-1 kinase is inhibited both in tumor cells and T cells). If the postulation is correct, the antitumor activity in (R)-9b should be better than ACK1-KO mouse model. Nonetheless, in the manuscript, there is no data to confirm that ablation of ACK1 could induce the TME more attractive to T cells. Thus, the title of “Parallel Host T Cell Reinvigoration and Augmentation of Leukocyte Attractant by Cancer Cell Upon ACK1 inhibition Overcomes Immune Resistance” is not suitable to this study.

Response: As per suggestion, we have changed title, the new title is:

T Cell Reinvigoration upon ACK1/TNK2 inhibition Overcomes Immune Resistance

3. In the experiments of Fig. 4, 5, 7, TIL (tumor infiltrating lymphocytes) analysis by flowcytometry or immunohistochemical study of tumor tissues are needed to evaluate the TME.

Response: The analysis of the TILs has been done, either by flow cytometry or IHC, and the data has been included in this revised manuscript. Please see **Supplementary Fig. 8a, 8c, 13e, 16a, 16b.**

Minor issues:

The data of tumor weight Fig. 4b, 5b and 7c are not necessary, they are overlapped with tumor volume data Fig. 4a, 5a, and 7a.

Response: As suggested by the reviewer, the data on tumor weights have been removed from the figures 4, 5 and 7 and has been placed in the **Supplementary Fig. 7a, 9b, 13b, 13c and 14d.**

REVIEWER COMMENTS

Reviewer #2 (Remarks to the Author):

The authors have addressed my concerns and I approve the manuscript publication.

Reviewer #3 (Remarks to the Author):

In this revised manuscript, the authors adequately addressed all comments raised by the reviewers.

Reviewer #4 (Remarks to the Author):

The authors have made efforts to respond to specific comments by reviewers well, but unfortunately, they have not responded in an adequate way and in addition, the revised manuscript lost the important scenario of their paper. I think that they should reorganize the text, Figs and Tables, including supplemental materials. The data presented are so redundant. They need to select the required and informative data.

Specific Points

1) There are still so many inadequate descriptions as a science paper. The problems are not simple grammatical errors. The authors often used emotional words, words implying personal thinking in the abstract and introduction, and robust descriptions without explanation. For examples, in the Abstract, back to/CSK's/critical/interestingly/self-sabotaging loop/harnessing unique dichotomous mode/unprecedented opportunity; In the Introduction, that poses a significant challenge due to --- -/ chronic tumor antigen stimulation leading to ---/ Interestingly, -- too---/ Intrigued by ---/ crucial events/ Serendipitously, / back to / in immunologically cold tumors. I think that the authors need to revise their description throughout the text.

2) They should reorganize the description of Page 4 in more simple way. I think that the reviewer did not request to show so many complicated supplemental data including Suppl Table 1. In addition, why they need to show Sup Fig. 2a and Sup Fig. 3a? How did they identify MDSC? This data is from intact mice. They should not use MDSC without definition. They should simply describe the CD marker for phenotyping. Delete Supple Table 1. It's not informative Regarding to Fig. 1d/e and Sup Fig 2 and 3 description, they should change to " The percentages of CD44^{hi}CD62L^{low} memory CD8⁺ T cells and CD137⁺ effector CD8⁺ T cells increased in the spleen of ack1 KO mice, albeit no differences were observed in the ratios of Foxp3-CD4⁺ helper T, Foxp3+CD4⁺ regulatory T (Treg), CD8⁺ CTL, CD2⁺ B, CD2⁺ NK, CD2⁺ monocytes and CD11b⁺Gr-1⁺ myeloid cells. In Suppl Fig. 2C figure legend, they need to state how they defined immune subsets.

3) The reviewer only requested the explanation. They did not respond in an adequate way. The addition of so many blind supplemental data and description in the results broken the flow (main story) of the paper. Delete Supple Table 2.

4) I think that the original reviewer just requested to delete thymus data. Or do they want to mention all thymocyte and peripheral T cells are spontaneously activated in KO mice? They need to demonstrate more clearly.

In addition, they do not need to show all results that they did. Please delete Fig. 1f (and all thymus data hereafter) and related description. In the present scenario, they do not need to show thymocyte data.

5) They have not adequately response to the comment. They need to describe the reason to examine Ca⁺⁺ influx, the meaning of addition of anti-CD3 mAb and ionomycin in the result, and their doses in the legend.

6) Why they added the data of Jurkat cells? The reviewer did not request it.

Point-by-point response to Reviewer's Comment

Reviewer #2 (Remarks to the Author):

The authors have addressed my concerns and I approve the manuscript publication.

Response: We thank the reviewer.

Reviewer #3 (Remarks to the Author):

In this revised manuscript, the authors adequately addressed all comments raised by the reviewers.

Response: We thank the reviewer.

Reviewer #4 (Remarks to the Author):

The authors have made efforts to respond to specific comments by reviewers well, but unfortunately, they have not responded in an adequate way and in addition, the revised manuscript lost the important scenario of their paper. I think that they should reorganize the text, Figs and Tables, including supplemental materials. The data presented are so redundant. They need to select the required and informative data.

Response: We have made all the edits as per reviewer's suggestions, as described below.

Specific Points

1) There are still so many inadequate descriptions as a science paper. The problems are not simple grammatical errors. The authors often used emotional words, words implying personal thinking in the abstract and introduction, and robust descriptions without explanation. For examples, in the Abstract, back to/CSK's/critical/interestingly/self-sabotaging loop/harnessing unique dichotomous mode/unprecedented opportunity; In the Introduction, that poses a significant challenge due to ----/ chronic tumor antigen stimulation leading to ---/ Interestingly, -- too---/ Intrigued by ---/ crucial events/ Serendipitously, / back to / in immunologically cold tumors. I think that the authors need to revise their description throughout the text.

Response: We have made all the edits as per reviewer's suggestions throughout the manuscript.

2) They should reorganize the description of Page 4 in more simple way. I think that the reviewer did not request to show so many complicated supplemental data including Suppl Table 1. In addition, why they need to show Sup Fig. 2a and Sup Fig. 3a?

Response: We have deleted Sup Fig. 2a and Sup Fig. 3a.

How did they identify MDSC? This data is from intact mice. They should not use MDSC without definition. They should simply describe the CD marker for phenotyping.

Response: As per suggestion, we have marked the cells as CD11b⁺ Gr1⁺. Other markers too are described.

Delete Supple Table 1.

Response: We have deleted Supple Table 1.

Regarding to Fig. 1d/e and Sup Fig 2 and 3 description, they should change to “ The percentages of CD44^{hi}CD62L^{low} memory CD8⁺ T cells and CD137⁺ effector CD8⁺ T cells increased in the spleen of ack1 KO mice, albeit no differences were observed in the ratios of Foxp3-CD4⁺ helper

T, Foxp3+CD4+ regulatory T (Treg), CD8+ CTL, CD4+ B, CD4+ NK, CD4+ monocytes and CD11b+Gr-1+ myeloid cells.

Response: This sentence has been incorporated, replacing earlier version.

In Suppl Fig. 2C figure legend, they need to state how they defined immune subsets.

Response: The figure legend has been corrected as per suggestion.

3) The reviewer only requested the explanation. They did not respond in an adequate way. The addition of so many blind supplemental data and description in the results broken the flow (main story) of the paper. Delete Supple Table 2.

Response: We have checked IL4 levels and they were not affected. We have deleted Supple Table 2.

We have no evidence to suggest any relationship between AKT and IFN- γ production in *Ack1/Tnk2* KO mice or (*R*)-9b treated mice. Human CD8+ T cells can be divided into IFN- γ - and IL-2-producing cells (Benoît P. Nicolet et. al. *PNAS*, 2020; Vrbensky JR, et. al. *Br J Haematol.* 2020). The unbiased transcriptomics and proteomics analysis on CD8+ T cells revealed that IFN- γ + cells produce cytotoxic molecules. During infection and cancers, antigen-specific immunity drives the production of IFN- γ by CD8+ T lymphocytes (Abd Hamid M. et., al. *Front Immunol.* 2020; 5;11:310). Taken together these data explain increased IFN- γ levels in KO mice and (*R*)-9b treated mice, due to CD8 T-cell activation.

4) I think that the original reviewer just requested to delete thymus data. Or do they want to mention all thymocyte and peripheral T cells are spontaneously activated in KO mice? They need to demonstrate more clearly. In addition, they do not need to show all results that they did. Please delete Fig. 1f (and all thymus data hereafter) and related description. In the present scenario, they do not need to show thymocyte data.

Response: We have deleted all the thymocyte data.

5) They have not adequately response to the comment. They need to describe the reason to examine Ca⁺⁺ influx, the meaning of addition of anti-CD3 mAb and ionomycin in the result, and their doses in the legend.

Response: Calcium influx is considered as one of the crucial events during T cell receptor (TCR) engagement and subsequent signaling for the T cell activation and has been used regularly (Lo, et al., 2018; Christo SN et al., 2015, Joseph N et al., 2014, Thaker, Y. R. et al. 2017; please see below). The calcium flux changes were observed upon stimulation of splenocytes by addition of anti-CD3 antibody (concentration of 0.5 μ g/mL) and calcium overload (saturation) by addition of ionomycin (concentration of 1 μ g/ml), which is known to induce a saturating, sustained increase in calcium ions (Morgan AJ and Jacob R, 1994).

We have added this information as well as references in the manuscript.

References

- Lo WL, Shah NH, Ahsan N, Horkova V, Stepanek O, Salomon AR, Kuriyan J, Weiss A. Lck promotes Zap70-dependent LAT phosphorylation by bridging Zap70 to LAT. *Nature Immunol.* 2018 Jul;19(7):733-741. doi: 10.1038/s41590-018-0131-1. Epub 2018 Jun 18. PMID: 29915297; PMCID: PMC6202249.
- Joseph N, Reicher B, Barda-Saad M. The calcium feedback loop and T cell activation: how cytoskeleton networks control intracellular calcium flux. *Biochim Biophys Acta.*

2014 Feb;1838(2):557-68. doi: 10.1016/j.bbamem.2013.07.009. Epub 2013 Jul 13. PMID: 23860253.

- Christo SN, Diener KR, Nordon RE, Brown MP, Griesser HJ, Vasilev K, Christo FC, Hayball JD. Scrutinizing calcium flux oscillations in T lymphocytes to deduce the strength of stimulus. *Sci Rep*. 2015 Jan 14;5:7760. doi: 10.1038/srep07760. PMID: 25585590; PMCID: PMC4293621.
- Thaker, Y. R. et al. Activated Cdc42-associated kinase 1 (ACK1) binds the sterile alpha motif (SAM) domain of the adaptor SLP-76 and phosphorylates proximal tyrosines. *J Biol Chem* 292, 6281-6290, doi:10.1074/jbc.M116.759555 (2017).
- Morgan AJ, Jacob R. Ionomycin enhances Ca²⁺ influx by stimulating store-regulated cation entry and not by a direct action at the plasma membrane. *Biochem J*. 1994 Jun 15;300 (Pt 3)(Pt 3):665-72. doi: 10.1042/bj3000665. PMID: 8010948; PMCID: PMC1138219.

6) Why they added the data of Jurkat cells? The reviewer did not request it.

Response: We have deleted the Jurkat data.

REVIEWERS' COMMENTS

Reviewer #4 (Remarks to the Author):

The authors adequately responded to the comments in the revised manuscript.
The flow of manuscript has been improved now.

Minor comments

Fig. 1 h and figure legend; Ifn should change to IFN.

ELISA section in the Methods; Interferon- α should change to γ .

Point-by-point response to Reviewer's Comment

Reviewer #4 (Remarks to the Author):

The authors adequately responded to the comments in the revised manuscript. The flow of manuscript has been improved now.

Response: We thank the reviewer.

Minor comments

Fig. 1 h and figure legend; Ifn should change to IFN. ELISA section in the Methods; Interferon- α should change to γ .

Response: In Fig. 1h and legend, Ifn has been changed to IFN. In ELISA methodology, Interferon- α has been corrected as γ .